# Transient protein accumulation at the center of the T cell antigen-presenting cell interface drives efficient IL-2 secretion

Danielle J Clark[1†], Laura E McMillan[1†], Sin Lih Tan[1], Gaia Bellomo[1], Clementine Massoue[1], Harry Thompson[1], Lidiya Mykhaylechko[1], Dominic Alibhai[2], Xiongtao Ruan[3], Kentner L Singleton[4], Minna Du[4], Alan Hedges[1], Pamela L Schwartzberg[5], Paul Verkade[2], Robert F Murphy[3,6,7,8,9,10], Christoph Wülfing[1,4]*

[1]School of Cellular and Molecular Medicine, University of Bristol, Bristol, United Kingdom; [2]School of Biochemistry, University of Bristol, Bristol, United Kingdom; [3]Computational Biology Department, School of Computer Science, Carnegie Mellon University, Pittsburgh, United States; [4]Department of Immunology, University of Texas Southwestern Medical Center, Dallas, United States; [5]Genetic Disease Research Branch, National Human Genome Research Institute, National Institutes of Health, Bethesda, United States; [6]Department of Biological Sciences, Carnegie Mellon University, Pittsburgh, United States; [7]Department of Biomedical Engineering, Carnegie Mellon University, Pittsburgh, United States; [8]Department of Machine Learning, Carnegie Mellon University, Pittsburgh, United States; [9]Freiburg Institute for Advanced Studies, Albert Ludwig University of Freiburg, Freiburg, Germany; [10]Faculty of Biology, Albert Ludwig University of Freiburg, Freiburg, Germany

*For correspondence:
Christoph.Wuelfing@bristol.ac.uk

[†]These authors contributed equally to this work

Competing interests: The authors declare that no competing interests exist.

**Abstract** Supramolecular signaling assemblies are of interest for their unique signaling properties. A µm scale signaling assembly, the central supramolecular signaling cluster (cSMAC), forms at the center of the interface of T cells activated by antigen-presenting cells. We have determined that it is composed of multiple complexes of a supramolecular volume of up to 0.5 µm$^3$ and associated with extensive membrane undulations. To determine cSMAC function, we have systematically manipulated the localization of three adaptor proteins, LAT, SLP-76, and Grb2. cSMAC localization varied between the adaptors and was diminished upon blockade of the costimulatory receptor CD28 and deficiency of the signal amplifying kinase Itk. Reconstitution of cSMAC localization restored IL-2 secretion which is a key T cell effector function as dependent on reconstitution dynamics. Our data suggest that the cSMAC enhances early signaling by facilitating signaling interactions and attenuates signaling thereafter through sequestration of a more limited set of signaling intermediates.

## Introduction

T cell activation is governed by spatiotemporal organization of signal transduction across scales. At the nanoscale, receptors form clusters of dozens of molecules that can coalesce into microclusters and are commonly associated with active forms of signaling intermediates (*Boyle et al., 2011*; *Hu et al., 2016*; *Lillemeier et al., 2006*; *Schamel et al., 2005*; *Sherman et al., 2011*; *Varma et al., 2006*; *Yokosuka et al., 2005*). This association suggests that such receptor clusters mediate efficient T cell signaling. Larger, µm scale assemblies were first described at the center and periphery of T

**eLife digest** Cells receive dozens of signals at different times and in different places. Integrating incoming information and deciding how to respond is no easy task. Signaling molecules on the cell surface pass messages inwards using chemical messengers that interact in complicated networks within the cell. One way to unravel the complexity of these networks is to look at specific groups of signaling molecules in test tubes to see how they interact. But the interior of a living cell is a very different environment. Molecules inside cells are tightly packed and, under certain conditions, they interact with each other by the thousands. They form structures known as 'supramolecular complexes', which changes their behavior.

One such supramolecular complex is the 'central supramolecular activation cluster', or cSMAC for short. It forms under the surface of immune cells called T cells when they are getting ready to fight an infection. Under the microscope, the cSMAC looks like the bullseye of a dartboard, forming a crowd of signaling molecules at the center of the interface between the T cell and another cell. Its exact role is not clear, but evidence suggests it helps to start and stop the signals that switch T cells on. The cSMAC contains two key protein adaptors called LAT and SLP-76 that help to hold the structure together. So, to find out what the cSMAC does, Clark et al. genetically modified these adaptors to gain control over when the cSMAC forms.

Clark et al. examined mouse T cells using super-resolution microscopy and electron microscopy, watching as other immune cells delivered the signal to switch on. As the T cells started to activate, the composition of the cSMAC changed. In the first two minutes after the cells started activating, the cSMAC included a large number of different components. This made T cell activation more efficient, possibly because the supramolecular complex was helping the network of signals to interact. Later, the cSMAC started to lose many of these components. Separating components may have helped to stop the activation signals.

Understanding how T cells activate could lead to the possibility of turning them on or off in immune-related diseases. But these findings are not just relevant to immune cells. Other cells also use supramolecular complexes to control their signaling. Investigating how these complexes change over time could help us to understand how other cell types make decisions.

cells activated by antigen-presenting cells (APC) for the TCR, PKCθ and LFA-1, talin, respectively, as central and peripheral supramolecular activation clusters (cSMAC and pSMAC) (*Grakoui et al., 1999*; *Monks et al., 1998*; *Monks et al., 1997*). μm scale of assemblies, in particular in the form of supramolecular protein complexes, provides unique biophysical and signaling properties (*Banani et al., 2016*; *Li et al., 2012*; *Shin and Brangwynne, 2017*). Supramolecular protein complexes play critical roles in viral sensing (*Cai et al., 2014*), inflammation (*Franklin et al., 2014*), embryonic development (*Brangwynne et al., 2009*), protein folding in cancer (*Rodina et al., 2016*), nuclear ubiquitinoylation (*Marzahn et al., 2016*), and chromatin compaction (*Larson et al., 2017*). Such complexes are readily observed by fluorescence microscopy, held together by a network of multivalent protein interactions and often have distinct phase properties (*Banani et al., 2016*; *Li et al., 2012*; *Shin and Brangwynne, 2017*). The cSMAC has many properties of such supramolecular protein complexes: It contains various multivalent signaling intermediates (*Balagopalan et al., 2015*), prominently LAT (linker of activation of T cells), components of this complex including LAT and PKCθ exchange with the remainder of the cell to a moderate extent and slowly (*Roybal et al., 2015*), and components of this complex can be assembled into supramolecular structures in vitro (*Su et al., 2016*). Therefore, understanding biophysical properties of the cSMAC and how it regulates T cell activation is of substantial importance. cSMAC function is controversial despite decades of work. Limitations in investigating the cSMAC are that its properties are largely unresolved and that its composition and/or assembly have not been systematically manipulated inside live T cells. Based on association of cSMAC formation with T cell activation conditions, the cSMAC has been proposed to enhance T cell signaling, terminate it, not be related to signaling or only upon weak stimulation or at late time points (*Čemerski et al., 2008*; *Freiberg et al., 2002*; *Grakoui et al., 1999*; *Lee et al., 2002*; *Monks et al., 1997*). cSMAC formation is often associated with efficient T cell activation conditions, fitting with a role in enhancing T cell signaling. Accumulation of signaling

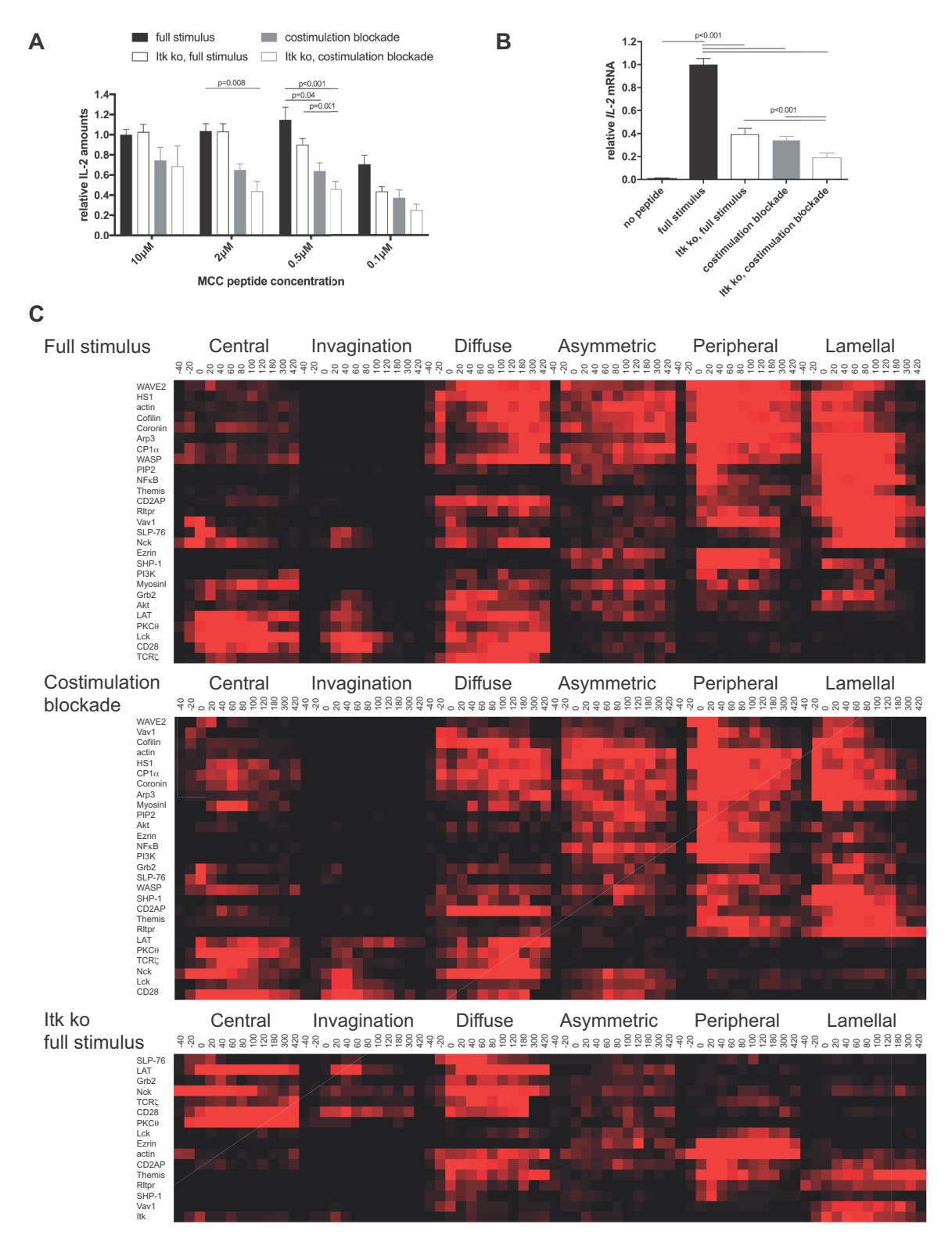

**Figure 1.** CD28 and Itk regulate IL-2 secretion and signaling organization. (**A**) In vitro primed 5C.C7 T cells, wild type or Itk-deficient ('Itk ko'), were activated by CH27 APCs and the indicated concentration of MCC *pept*ide in the absence or presence of 10 μg/ml anti-CD80 plus anti-CD86 ('full stimulus' or 'costimulation blockade'). IL-2 levels in the supernatant are given relative to stimulation of wild type 5C.C7 T cells under full stimulus conditions with 10 μM MCC with SEM. 4–8 experiments were averaged per condition. Statistical significance as determined separately for each MCC

*Figure 1 continued on next page*

*Figure 1 continued*

peptide concentration by 1-way ANOVA is indicated. (**B**) Relative levels of *IL-2* mRNA are given upon 5C.C7 T cell activation similar to A with only 10 µM MCC. 3–18 experiments were averaged per condition. Statistical significance as determined by 1-way ANOVA is indicated. (**C**) Wild type and Itk-deficient ('Itk ko') 5C.C7 T cells expressing the indicated sensors were activated by CH27 B cell APCs (10 µM MCC) in the absence or presence of 10 µg/ml anti-CD80 plus anti-CD86 (('full stimulus' or 'costimulation blockade') and percentage occurrence of patterns of interface enrichment (***Figure 1—figure supplement 2A***) is given in shades of red from −40 to 420 s relative to tight cell coupling. Cluster trees are given in pink. Sensors used and source data for panel Care given in ***Figure 1—figure supplements 2B–4***, ***Figure 1—source data 1***.

The online version of this article includes the following source data and figure supplement(s) for figure 1:

**Source data 1.** Sensors used in ***Figure 1C,D***.
**Figure supplement 1.** The generation of *IL-2* mRNA is focused on the first 6 hr after cell couple formation.
**Figure supplement 2.** Sensor accumulation at the T cell:APC interface under full stimulus conditions.
**Figure supplement 3.** Sensor accumulation at the T cell:APC interface under costimulation blocked conditions.
**Figure supplement 4.** Sensor accumulation at the T cell:APC interface in the absence of Itk.

intermediates at the T cell:APC interface center is substantially reduced by blockade of the costimulatory receptor CD28 (*Singleton et al., 2009*; *Wülfing et al., 2002*), in regulatory T cells (*Zanin-Zhorov et al., 2010*), during thymic selection (*Ebert et al., 2008*), or in the absence of the signal amplifying kinase Itk (IL-2 inducible T cell kinase) (*Singleton et al., 2011*). To determine cSMAC properties, we have used stimulated emission depletion (STED) super-resolution microscopy and correlative light electron microscopy (CLEM). The cSMAC was composed of multiple complexes of supramolecular dimensions and associated with extensive membrane undulations. To investigate cSMAC function, we have systematically manipulated the localization of three adaptor proteins in live primary T cells: LAT (*Balagopalan et al., 2015*) is an integral component of the cSMAC. SLP-76 (SH2 domain-containing leucocyte protein of 76 kD) (*Koretzky et al., 2006*) is associated with it only during the first minute of T cell activation (*Roybal et al., 2015*). Grb2 (growth factor receptor-bound 2) (*Jang et al., 2009*) association with the cSMAC is less prevalent (*Roybal et al., 2015*). Interface recruitment of all three adaptors was diminished upon attenuation of T cell activation by costimulation blockade and Itk-deficiency as was IL-2 secretion, a critical T cell effector function. By fusing these adaptors with various protein domains with a strong interface localization preference we brought them back to the interface under the attenuated T cell activation conditions and restored cSMAC formation. Such restoration enhanced IL-2 secretion but only when executed to the extent and with dynamics seen under full stimulus conditions.

## Results

### µm scale LAT accumulation at the center of the T cell APC interface is associated with efficient T cell activation

To investigate the function of µm scale protein accumulation at the center of the T cell:APC interface, we first cataloged protein localization events that were consistently associated with efficient T cell activation. We attenuated T cell activation through costimulation blockade (*Singleton et al., 2009*; *Wülfing et al., 2002*) and Itk deficiency. The 5C.C7 T cell receptor (TCR) recognizes the moth cytochrome C (MCC) 89–103 peptide presented by I-E$^k$. In the restimulation of in vitro primed 5C.C7 T cells with CH27 B cell lymphoma APCs and MCC peptide IL-2 amounts in the supernatant were reduced upon blockade of the CD28 ligands CD80 and CD86 ('costimulation blockade') and in T cells from Itk knock out 5C.C7 TCR transgenic mice, in particular at lower peptide concentrations (***Figure 1A***). As IL-2 amounts in T cell culture supernatants are determined by the difference between IL-2 generation and consumption we also determined *IL-2* mRNA levels. Even at an MCC peptide concentration of 10 µM the level of *IL-2* mRNA in T cells was significantly ($p < 0.001$) reduced to less than 50% upon costimulation blockade and Itk-deficiency (***Figure 1B***). 10 µM MCC was used for the remainder of the study. To more precisely relate the determination of IL-2 amounts in T cell culture supernatants to *IL-2* mRNA generation, we determined the time course of both (***Figure 1—figure supplement 1***). *IL-2* mRNA generation occurred during the first six hours of T cell activation, consistent with transient nuclear localization of NFkB and previous data establishing that APC contact times of less than one hour are sufficient to commit a primed T cells to proliferation (*Iezzi et al.,*

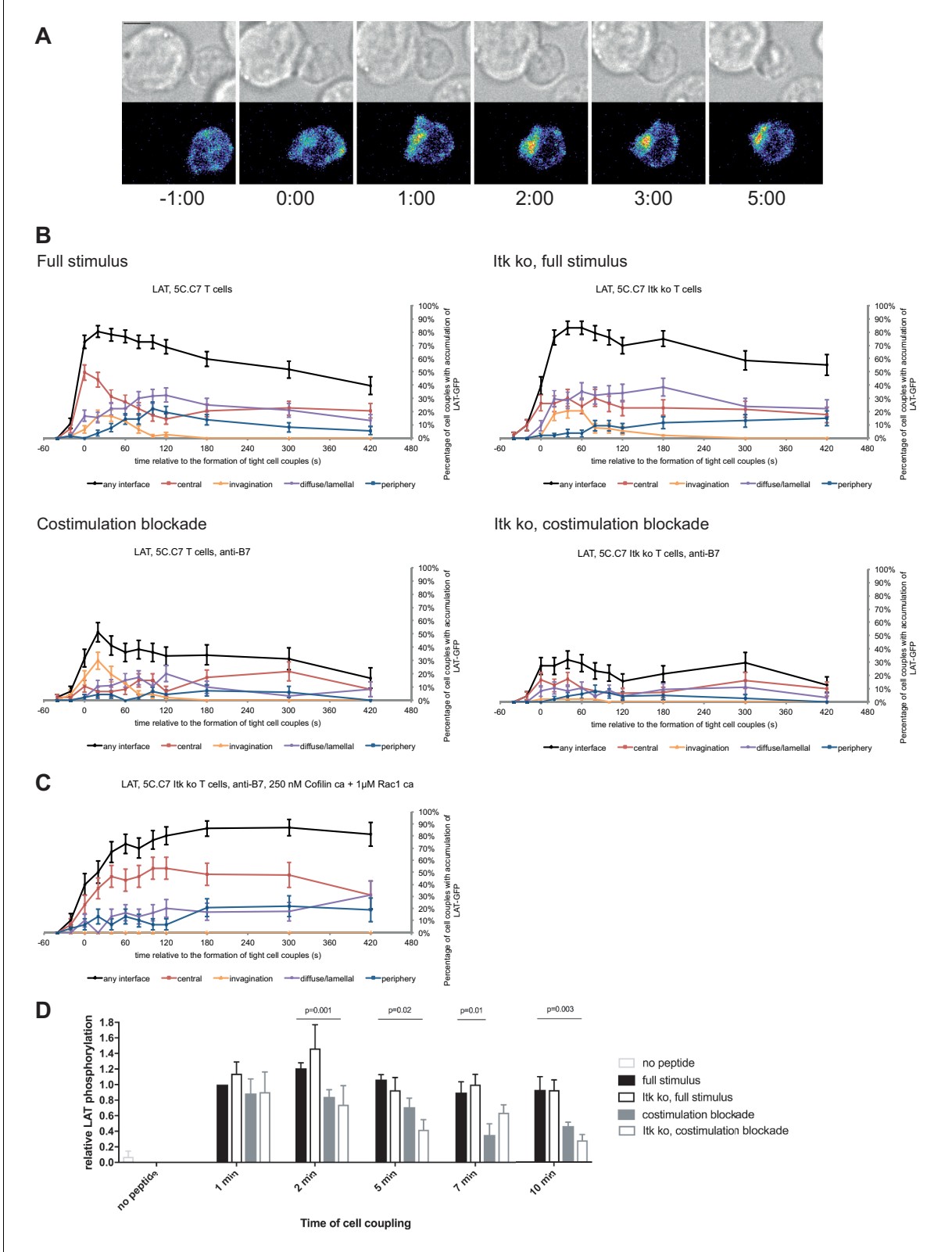

**Figure 2.** LAT localization and activation is regulated by costimulation and Itk. (**A**) An interaction of a LAT-GFP-transduced 5C.C7 T cell with a CH27 APC (10 μM MCC) is shown at the indicated time points (in minutes) relative to the time of formation of a tight cell couple. Differential interference contrast (DIC) images are shown in the top row, with top-down, maximum projections of 3-dimensional LAT-GFP fluorescence data in the bottom row. LAT-GFP fluorescence intensities are displayed in a rainbow-like false-color scale (increasing from blue to red). The scale bar corresponds to 5 μm. A

*Figure 2 continued on next page*

*Figure 2 continued*

corresponding video is available as *Figure 2—Video 1*. (B) The graphs display the percentage of cell couples with LAT accumulation in the indicated patterns (*Figure 1—figure supplement 2A*, 'periphery' is the sum of asymmetric and peripheral) relative to tight cell couple formation for wild type or Itk-deficient 5C.C7 T cells activated with CH27 APCs (10 µM MCC) in the absence or presence of 10 µg/ml anti-CD80 plus anti-CD86 ('costimulation blockade') as indicated. 47–77 cell couples from 2 to 5 independent experiments were analyzed per condition, 226 total. A statistical analysis is given in *Figure 2—source data 1*. (C) Itk-deficient 5C.C7 T cells were activated with CH27 APCs (10 µM MCC) in the presence of 10 µg/ml anti-CD80 plus anti-CD86 and 250 nM constitutively active Cofilin plus 1 µM constitutively active Rac1 as protein transduction reagents. LAT interface accumulation is given as in B. 30 cell couples from a single experiment were analyzed. Statistical significance is given in *Figure 2—source data 1*. (D) 5C.C7 T cells were activated as in B for the indicated times. Band intensities of α-LAT pY191 blots (*Figure 2—figure supplement 1*) as normalized to the 1 min time point under full stimulus conditions are given. 5–7 experiments were averaged per condition. Statistical significance as determined separately for each time point by 1-way ANOVA is indicated.

The online version of this article includes the following video, source data, and figure supplement(s) for figure 2:

**Source data 1.** Statistical significance of differences in LAT accumulation under different T cell activation conditions is given for the indicated patterns as determined by proportion's z-test.

**Figure supplement 1.** Representative phospho-LAT Y191 western blots.

**Figure 2—video 1.** Representative interactions of 5C. C7 T cells retrovirally transduced to express the indicated GFP fusion proteins with CH27 B cell lymphoma APCs and 10 µM MCC peptide in the presence or absence of 10 µg/ml anti-CD80 plus anti-CD86 ('costimulation blockade') are shown in *Figure 2—Video 1*, *Figure 4—Videos 1–3*, *Figure 7—Videos 1–3* and *Figure 8—Video 1*.

https://elifesciences.org/articles/45789#fig2video1

---

*1998*). We used *IL-2* mRNA generation for the remainder of the study because of its greater sensitivity to stimulus attenuation.

We characterized T cell signaling organization as extensively described before (*Ambler et al., 2017*; *Roybal et al., 2015*; *Singleton et al., 2009*). Briefly, in vitro primed 5C.C7 T cells are retrovirally transduced to express fluorescent signaling intermediates or sensors, FACS sorted to low expression as close as possible to endogenous signaling intermediate concentrations and imaged in three dimensions over time during restimulation with APC and 10 µM MCC peptide ('full stimulus'). In image analysis the frequency of occurrence of geometrically quantified µm scale subcellular distributions that represent underlying cell biological structures is determined (*Figure 1—figure supplement 2A*) (*Roybal et al., 2013*). Of particular interest here are accumulation at the center of the T cell APC interface ('central'), the cSMAC, and accumulation in a µm deep 'invagination' at the center of the interface that likely mediates termination of early central signaling (*Singleton et al., 2006*). Upon costimulation blockade most sensors that displayed frequent central accumulation upon full T cell stimulation, in particular during the first two minutes of cell coupling, did less so. In Itk-deficient 5C.C7 T cells sensors with only transient early central accumulation in wild type T cells lost much of that accumulation (*Figure 1C,D*; *Figure 1—figure supplements 2B–4*, *Figure 1—source data 1*). Both phenotypes are indicative of reduced cSMAC formation. Efficient IL-2 secretion thus was associated with µm scale central signaling localization.

LAT as a key cSMAC component displayed µm scale central localization (*Figure 2A*) in a biphasic pattern. At the time of tight cell coupling under full stimulus conditions 49 ± 6% of cell couples showed central LAT accumulation. After 2 min of cell coupling central LAT accumulation was only found in about 25% of cell couples (*Figure 2B*) and remained stable at that level. In the absence of Itk, the initial peak of central LAT accumulation was significantly (p=0.005) diminished to 26 ± 6% of cell couples with central LAT accumulation. Such reduction was more pronounced (11 ± 4%, p<0.001) upon costimulation blockade (*Figure 2B*; *Figure 2—source data 1*). Combining costimulation blockade and Itk deficiency yielded the least interface LAT accumulation (*Figure 2B*). Impaired activation of cytoskeletal transport processes is a likely contributor to diminished central LAT accumulation upon costimulation blockade and Itk deficiency, as enhancement of actin dynamics with active Rac and Cofilin (*Roybal et al., 2016*) significantly (p<0.001, *Figure 2C*; *Figure 2—source data 1*)(*Roybal et al., 2016*) increased central and overall LAT accumulation.

Attenuation of T cell activation was associated with diminished LAT phosphorylation at Y191 (*Figure 2D*; *Figure 2—source data 1*) upon costimulation blockade, in particular in combination with Itk deficiency. At 2, 5, and 10 min after tight cell coupling LAT phosphorylation was significantly (p≤0.02) reduced in Itk-deficient 5C.C7 T cells upon costimulation blockade compared to wild type 5C.C7 T cells under full stimulus conditions by 39%, 61%, and 70%, respectively.

## The cSMAC consists of multiple complexes of supramolecular dimensions and is associated with extensive membrane undulations

To determine cSMAC properties, we stained 5C.C7 T cell:CH27 B cell APC couples for LAT and LAT phosphorylated at Y191 ('pLAT') and imaged them using STED super-resolution microscopy (*Figure 3A*). Multiple LAT/pLAT complexes formed in the cSMAC and/or beyond, on average four per cell. LAT complexes were significantly larger (p=0.04) in the cSMAC region with a supramolecular volume of $0.23 \pm 0.03$ μm$^3$ than in T cells without a cSMAC ($0.12 \pm 0.01$ μm$^3$)(*Figure 3B*). pLAT complexes were similarly larger in the cSMAC region with volumes of $0.25 \pm 0.03$ μm$^3$ (cSMAC) versus $0.13 \pm 0.01$ μm$^3$ (non-cSMAC, p=0.007)(*Figure 3B*).

To understand how LAT as a transmembrane protein could drive the formation of multiple supramolecular complexes, we related cSMAC formation to plasma membrane topology with CLEM (*Figure 3C*; *Figure 3—figure supplement 1*). During live cell imaging of 5C.C7 T cells activated by CH27 B cell APCs we rapidly fixed samples upon detection of central LAT clustering and processed them for electron microscopy (EM). We used two experimental conditions, a full stimulus and activation of 5C.C7 T cells expressing a fusion protein of LAT with the PKCθ V3 domain upon costimulation blockade as an experimental strategy to achieve enhanced cSMAC formation upon attenuation of T cell activation, as introduced in detail in the next section. In EM sections we measured the extent of membrane undulations in cSMAC and non-cSMAC interface regions as the ratio of the plasma membrane length to the straight-line diameter of the region. This ratio was significantly (p<0.001) larger in the cSMAC ($2.2 \pm 0.3$) than in the pSMAC of the same T cell ($1.4 \pm 0.1$). In control cell couples without cSMAC formation the ratio as measured across the entire interface was as small ($1.5 \pm 0.1$) as that in pSMAC regions of cell couples with central LAT clustering (*Figure 3D*). The non-cSMAC data are consistent with diminished membrane undulations in 5C.C7 T cells upon costimulation blockade with a ratio of $1.5 \pm 0.1$ (*Roybal et al., 2016*).

The cSMAC thus is a cellular region where membrane undulations and multiple supramolecular complexes are associated with enhanced proximity of signaling intermediates. To enable the throughput required for functional cSMAC investigation in the remainder of these studies, we used spinning disk confocal microscopy to determine central protein accumulation as a measure of cSMAC formation.

## Fusion of LAT with protein domains with pronounced interface localization preference controls LAT localization

To determine cSMAC function, we wanted to restore its formation upon costimulation blockade and in Itk-deficient T cells. To do so, we needed to hypothesize how cSMAC components interact within the complex. Conceptually, a supramolecular complex could function rigidly through formation of stoichiometrically defined protein interactions or flexibly by enhancing signaling proximity through complex formation such that the same proteins can be assembled using varying stoichiometries and protein interaction motifs. Because of the large number of cSMAC components some of which are membrane-bound (*Figure 1*) we regard the flexible model as most likely. Impaired cSMAC formation upon costimulation blockade and Itk-deficiency should at least in part be driven by a reduction in the number of protein interaction motifs, that is the valence, of key components such as LAT as caused by diminished tyrosine phosphorylation (*Figure 2D*). We should therefore be able to restore cSMAC formation by enhancing valence through the addition of new protein interaction domains. While this involves slightly divergent stoichiometries and protein interaction motifs the live cell signaling functionality gained through the formation of the central region of increased protein density should be restored.

We increased LAT valence by adding three protein domains: PKCθ V3, Vav1 SH3SH2SH3, or PLCδ PH. The PKCθ V3 domain is required for central interface accumulation of PKCθ (*Kong et al., 2011*) even though it couldn't drive central localization by itself (*Figure 4—figure supplement 1A*). The Vav1 SH3SH2SH3 domains drove strong central accumulation only within the first minute of cell coupling (*Figure 4—figure supplement 1A*), as consistent with the localization of full length Vav1 (*Figure 1—figure supplement 2A*). The PLCδ PH domain mediated interface accumulation focused on the first two minutes of cell coupling without a central preference (*Figure 4—figure supplement 1A*). While the addition of protein interaction domains to LAT is predicted to alter their interactome, initial experiments to support this notion remained inconclusive as detailed in the Methods section.

Equal expression of LAT and the three LAT fusion proteins was enforced by FACS sorting for the same level of GFP. GFP fusion proteins were expressed at 2.1 ± 0.7 fold the endogenous level of LAT in non-transduced T cells (*Figure 4—figure supplement 1B,C*) with little change to endogenous LAT levels in the transduced T cells.

Fusion of LAT to the PKCθ V3 domain (LAT V3) yielded efficient central accumulation that was well sustained over the entire imaging time frame under all conditions at levels around 50% of cell couples with central accumulation (*Figure 4*). Starting 40 s after tight cell coupling such sustained central accumulation was significantly (p<0.05, *Figure 4—source data 1*) more frequent than central accumulation of non-targeted LAT under full stimulus at almost every time point. Fusion of LAT to the PKCθ V3 domain thus stabilized central LAT accumulation well beyond the levels seen for LAT alone under any physiological condition. Nevertheless, in cSMACs formed upon LAT V3 expression membrane undulations were similarly enhanced as in T cells expressing LAT (*Figure 3D*) supporting comparable cSMAC properties.

Fusion of LAT to the Vav1 SH3SH2SH3 domains ('LAT Vav') yielded different effects depending on the T cell activation conditions. Upon a full T cell stimulus LAT Vav resulted in diminished central and overall accumulation compared to LAT alone that was significant (p<0.05) across many time points (*Figure 4*; *Figure 4—source data 1*), suggesting that LAT Vav does not enhance properly assembled signaling complexes. Upon the three attenuated stimuli LAT Vav consistently enhanced central accumulation, most dramatically (p≤0.001, at most time points) for costimulation blockade in wild type and Itk deficient cells (*Figure 4*; *Figure 4—source data 1*). LAT Vav accumulation upon attenuated T cell stimulation in any pattern was largely indistinguishable from non-targeted LAT accumulation under full stimulus conditions (p>0.05, *Figure 4—source data 1*) and central accumulation was moderately enhanced only between 1 and 3 min after tight cell coupling. Fusion of LAT to the Vav1 SH3SH2SH3 domain thus allowed for fairly close reconstitution of full stimulus-type LAT localization upon attenuated T cell stimulation.

Fusion of LAT to the PLCδ PH domain ('LAT PLCδPH') resembled LAT Vav but was less powerful (*Figure 4*; *Figure 4—source data 1*). Upon the three attenuated T cell stimuli LAT PLCδPH moderately enhanced central and overall LAT accumulation but didn't consistently reach the same extent as LAT alone under full stimulus conditions. For example, in Itk-deficient 5C.C7 T cells upon costimulation blockade interface accumulation in any pattern was consistently enhanced (p<0.005 for time point 20 and later) from <32% to>55% upon expression of LAT PLCδPH. However, accumulation at the interface center only moderately increased from a range of 6–17% to 19–35%.

To ensure that overall interface accumulation of the targeted LAT constructs was comparable, we measured (*Ambler et al., 2017*; *Roybal et al., 2016*) their interface recruitment upon costimulation blocked conditions. All constructs showed substantial interface recruitment with moderately less LAT Vav recruitment at the last four time points (*Figure 4—figure supplement 1D,E*). To ensure functionality of the targeted LAT constructs, we showed that they were tyrosine phosphorylated upon T cell activation (*Figure 4—figure supplement 1F*).

Fusion of LAT with additional protein interaction domains thus allowed us to control central clustering: Fusion with the PKCθ V3 domain yielded consistently enhanced central localization, fusion with the Vav1 SH3SH2SH3 domains largely restored full stimulus-type LAT localization upon costimulation blockade and Itk deficiency and fusion with the PLCδ PH domain resulted in partial restoration.

## Restoration of LAT centrality yields enhanced *IL-2* mRNA production

To determine T cell function upon manipulation of LAT valence and localization, we measured *IL-2* mRNA induction upon 5C.C7 T cell activation with CH27 APCs and 10 μM MCC peptide, directly mirroring the imaging conditions. Expression of the targeting domains in isolation had only minor effects on *IL-2* mRNA amounts (*Figure 5—figure supplement 1*). Forcing exaggerated central LAT clustering by fusion with the PKCθ V3 domain did not affect *IL-2* mRNA amounts (*Figure 5A,B*) as further discussed below. In contrast, restoring LAT centrality under costimulation blocked and Itk deficient conditions to slightly higher (LAT Vav) or slightly lower (LAT PLCδPH) levels than seen for non-targeted LAT under full stimulus conditions yielded a consistent and largely significant (p<0.05) increase in *IL-2* mRNA (*Figure 5A,B*) to levels close to the amounts of *IL-2* mRNA in LAT-transduced 5C.C7 T cells under full stimulus conditions. For example, in LAT-expressing 5C.C7 T cells *IL-2* mRNA amounts dropped to 18 ± 6% and 18 ± 3% of full stimulus mRNA upon costimulation

blockade in wild type and Itk-deficient 5C.C7 T cells, respectively. Expression of LAT Vav restored *IL-2* mRNA to 41 ± 9% and 58 ± 10% (p<0.01), respectively, and expression of LAT PLCδPH to 50 ± 9% (p<0.05) and 88 ± 23%. µm scale central LAT interface accumulation thus supported efficient IL-2 secretion depending on accumulation extent and dynamics.

### Forcing central LAT localization only modestly enhances the central localization of related signaling intermediates

Next, we investigated with one example to which extent the forced relocalization of one signaling intermediate can drive analogous relocalization of others. We determined the subcellular distributions of Grb2, Lck and Vav1 in 5C.C7 T cells in the presence of LAT V3 using IRES-containing retroviral vectors for the parallel expression of GFP-tagged versions of the signaling intermediates alongside LAT V3. Under full stimulus conditions expression of LAT V3 moderately diminished interface recruitment of Grb2, Lck and Vav1 (*Figure 6*; *Figure 6—source data 1*) suggesting that excessive central LAT localization upsets a finely balanced signaling system. Upon costimulation blockade the localization of all three signaling intermediates was largely unaffected by LAT V3. Grb2 and Lck centrality was moderately enhanced in Itk-deficient 5C.C7 T cells. For example, while the percentage of Itk-deficient 5C.C7 T cells with central Grb2-GFP expression did not exceed 7% at 40 s after cell coupling and thereafter, upon co-expression of LAT V3 this percentage averaged 15% over the same time frame. Such moderate enhancement did not occur at the time of tight cell coupling but was restricted to later time points. In Itk-deficient 5C.C7 T cells upon costimulation blockade, however, Grb2 accumulation was substantially enhanced upon parallel expression of LAT V3, reaching levels of central Grb2 accumulation not seen under any other experimental condition investigated. The centrality of Vav1 as a signaling intermediate with only minor early central accumulation (*Figure 1—figure supplement 2B*) was not altered under any condition. The forced central localization of LAT thus could only draw in Grb2 and Lck as signaling intermediates with some intrinsic central localization preference to a mostly moderate extent and upon only some attenuated T cell stimuli.

### Restoration of SLP-76 centrality modestly enhances *IL-2* mRNA production

We investigated SLP-76 as an adaptor with more transient central accumulation. At the time of tight cell coupling under full stimulus conditions 45 ± 7% of the cell couples displayed SLP-76 accumulation at the interface center, similar to LAT. However, 80 s later this percentage dropped to less than 10% (*Figure 7A,B*). Also similar to LAT, the peak of central SLP-76 accumulation was significantly (p≤0.01) diminished upon costimulation blockade, Itk-deficiency and the combination of both to <27%,<15% and<11% of cell couples with central SLP-76 accumulation, respectively (*Figure 7B*; *Figure 7—source data 1*). To enhance SLP-76 valence and thus control its localization, we fused SLP-76 to the PKCθ V3 ('SLP-76 V3') or the Vav1 SH3SH2SH3 ('SLP-76 Vav') domain. Equal expression of SLP-76 and the two SLP-76 fusion proteins was enforced by FACS sorting for the same level of GFP. Both SLP-76 fusion constructs did not significantly affect SLP-76 centrality under full stimulus conditions, in wild type or Itk-deficient 5C.C7 T cells (*Figure 7B*; *Figure 7—source data 1*). However, accumulation of SLP-76 at the interface center was moderately but significantly (p<0.05 at at least two time points within the first minute of tight cell coupling, the peak of central SLP-76 accumulation) enhanced upon costimulation blockade in wild type and Itk-deficient 5C.C7 T cells reaching for example 67 ± 7% and 33 ± 7%, respectively, of cell couples with central SLP-76 accumulation upon expression of SLP-76 Vav (*Figure 7B*; *Figure 7—source data 1*). Interestingly, the enhancement of SLP-76 centrality was limited to the first minute of tight cell coupling, the time where non-targeted SLP-76 accumulated at the interface center.

Consistent with the enhancement of SLP-76 centrality we observed a modest increase in *IL-2* mRNA amounts under costimulation blocked conditions upon expression of SLP-76 V3 and SLP-76 Vav. Upon expression of non-targeted SLP-76 costimulation blockade in wild type and Itk-deficient 5C.C7 T cells reduced *IL-2* mRNA amounts to 45 ± 9% and 21 ± 1%, respectively, of full stimulus (*Figure 7C*). Expression of SLP-76 V3 restored *IL-2* mRNA amounts to 87 ± 20% and 74 ± 24%, respectively, expression of SLP-76 Vav to 73 ± 8% and 75 ± 30% without reaching statistical significance in the stringent 2-way ANOVA (*Figure 7C*). Importantly, enhancement of centrality and IL-2 secretion remained closely linked across multiple T cell activation conditions and spatially targeted

SLP-76 constructs (*Figure 5B*) thus corroborating the importance of µm scale central protein accumulation within the first two minutes of cell coupling for IL-2 secretion.

## PKCθ V3 and Vav1 SH3SH2SH3 don't affect Grb2 centrality and *IL-2* mRNA production

As a negative control we enhanced the valence of a signaling intermediate with more tentative central localization preference: We fused Grb2 to PKCθ V3 or Vav1 SH3SH2SH3. Equal expression of Grb2 and the two Grb2 fusion proteins was enforced by FACS sorting for the same level of GFP. Upon 5C.C7 T cell activation with a full stimulus Grb2 is efficiently recruited to the T cell:APC interface during the first two minutes of tight cell coupling, peaking at 80 ± 5% of cell couples with any interface accumulation. This overall interface accumulation was significantly (p≤0.02 at at least three time points within the first two minutes of tight cell coupling) diminished upon costimulation blockade, Itk-deficiency and both, remaining below 64%, 35% and 43%, respectively of cell couples with any interface accumulation (*Figure 8A,B*; *Figure 8—source data 1*). Distinguishing Grb2 from LAT and SLP-76, Grb2 accumulation at the interface center didn't exceed 20% under any of the T cell activation conditions. Fusion of Grb2 with PKCθ V3 ('Grb2 V3') or Vav1 SH3SH2SH3 ('Grb2 Vav') did not enhance centrality under any of the T cell activation conditions (*Figure 8B*, *Figure 8—source data 1*). Fusion with the PKCθ V3 domain did not substantially alter Grb2 localization at all (*Figure 8B*). Fusion with the Vav1 SH3SH2SH3 domain enhanced overall Grb2 interface recruitment across many time points under all T cell activation conditions (*Figure 8B*). However, most of this accumulation was in the peripheral pattern, a pattern common with full length Vav1 (*Figure 1—figure supplement 2B*). As an important negative control, a protein with a minor intrinsic central preference can thus not be forced to the interface center. Expression of Grb2 V3 or Grb2 Vav did not alter *IL-2* mRNA production under any of the T cell activation conditions (*Figure 8C*) despite the substantially enhanced overall interface accumulation upon expression of Grb2 Vav. These data thus provide an important specificity control for the selective functional importance of protein accumulation in the cSMAC.

## Discussion

The cSMAC was characterized by enhanced membrane undulations and the formation of multiple protein complexes of supramolecular dimensions in the 0.1–0.5 µm³ range. Supramolecular complexes can be driven by the polymerization of a single or few defined proteins (*Cai et al., 2014*; *Franklin et al., 2014*; *Li et al., 2012*; *Marzahn et al., 2016*) or, likely more common, they can consists of an agglomeration of large numbers of proteins (*Rodina et al., 2016*; *Tarantino et al., 2014*). While supramolecular complexes formed by a small number of components are often characterized by defined structures such as lipid droplets or fibers (*Shin and Brangwynne, 2017*), our understanding of supramolecular complexes built from a large number of components is limited. As close to half of the amount of three components of the central signaling complex, LAT, active Rac, and PKCθ, is immobile and the remainder exchanges only slowly with rest of the cell (*Roybal et al., 2015*) cSMAC signaling complexes likely have partial solid state properties. T cell signaling has been extensively characterized with super-molecular resolution in T cells activated with planar APC substitutes, supported lipid bilayers and antibody-coated cover slips. Similar to our findings large protein complexes, microclusters, were found. The microclusters are transported to the interface center to form a single µm scale structure (*Varma et al., 2006*; *Yokosuka et al., 2005*). However in contrast to T cell activation on planar surfaces, in T cell:APC couples membrane undulations form across the entire interface with F-actin structures perpendicular to the interface plane (*Roybal et al., 2015*) and thus impair rather than enhance centripetal transport. A large central invagination may remove proteins from the interface center (*Singleton et al., 2006*) and the central membrane undulations characterized here increase the number of transmembrane proteins in contact with a given cytoplasmic volume. Therefore, the cSMAC in T cell:APC couples will likely have similar molecular constituents as the central complex in T cells activated on planar substrates but distinct biophysical properties. It is the signaling functionality arising from these unique properties that we have investigated here. Nevertheless, the more straightforward access to high resolution imaging and synthetic system engineering using planar APC substitutes provides intriguing data that complement our findings. Supramolecular signaling complexes including LAT, SLP-76 and Grb2 have been reconstituted on

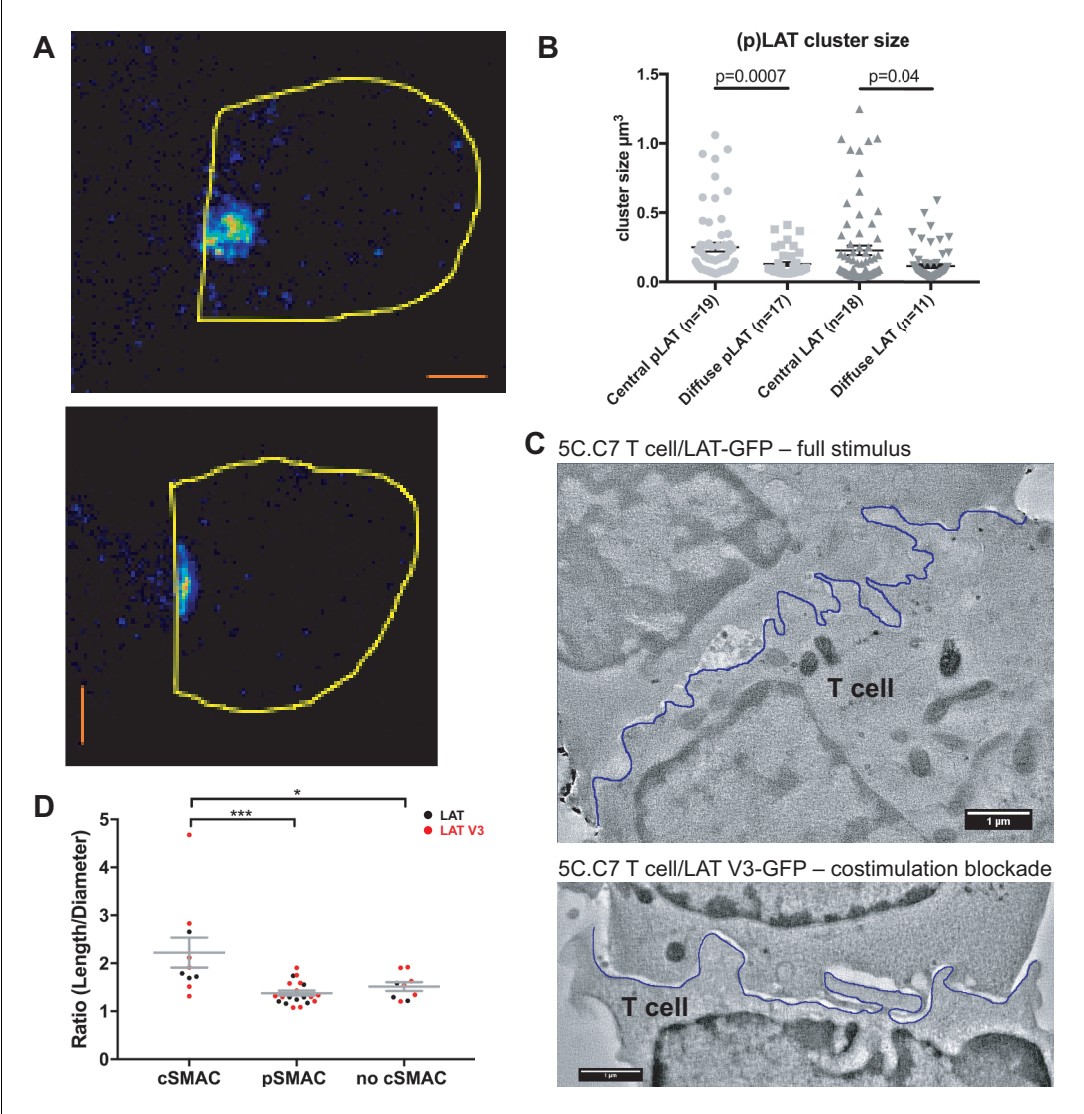

**Figure 3.** The cSMAC consists of multiple smaller complexes and is associated with extensive membrane undulations. (**A**) Two representative STED midplane images are given of 5C.C7 T cells activated by CH27 APCs (10 µm MCC) for 4.5 min and stained with α-LAT pY191. Staining fluorescence intensity is given in rainbow-like false-color scale (increasing from blue to red). The T cell outline is given in yellow. The scale bars correspond to 1 µm. (**B**) For experiments as in A LAT and LAT pY191 cluster size is given separately for cell couples with central or diffuse LAT accumulation as indicated (number of cell couples analyzed in two independent experiments in parentheses). Statistical significance as determined separately for 'LAT' and 'pLAT' by Student's test is indicated. (**C**) Midplane sections are given for two EM tomograms from a CLEM experiment. 5C.C7 T cells expressing LAT-GFP (top) or LAT V3-GFP (bottom)(see *Figure 4B*) were activated by CH27 APCs under full stimulus (top) or costimulation blocked (bottom) conditions, respectively. Upon formation of a cell couple with central LAT-GFP or LAT V3-GFP accumulation cell were fixed and processed for EM. The T cell plasma membrane at the cellular interface is traced in blue. Videos of the entire EM tomogram reconstructions are given as *Figure 3—Videos 1* and *2*. (**D**) In cell couples processed as in C membrane undulations were determined as the ratio of the length of the plasma membrane ('length') to a straight-line interface diameter of the same region ('diameter') in single images of EM sections. In cell couples with central LAT-GFP (black symbols) or LAT V3-GFP (red symbols) accumulation the interface center ('cSMAC') and periphery ('pSMAC') were measured separately. Peripheral regions were measured twice per cell, once to the left and once to the right of the central region. In control cell couples without central LAT-GFP (black symbols) or LAT V3-GFP (red symbols) accumulation ('no cSMAC') the entire interface was analyzed. Cell couples are derived from two independent experiments per condition. Statistical significance as determined by 1-way ANOVA is indicated (*p<0.05, ***p<0.001).

The online version of this article includes the following video and figure supplement(s) for figure 3:

**Figure supplement 1.** CLEM work flow.

**Figure 3—video 1.** Representative EM tomograms are shown in *Figure 3—Videos 1* and *2*.
https://elifesciences.org/articles/45789#fig3video1

**Figure 3—video 2.** The video is displayed similar to *Figure 3—Video 1*.

*Figure 3 continued on next page*

model membrane and shown to trigger signaling and F-actin assembly (*Su et al., 2016*). Large numbers of exosomes at the center of the interface between T cells and supported lipid bilayers could be related to the cSMAC membrane undulations and contribute to signal attenuation (*Choudhuri et al., 2014*). Highly localized F-actin structures discovered using T cell activation on planar surfaces could mediate transport into supramolecular signaling complexes (*Kumari et al., 2015*) with the F-actin uncapping protein RLTPR identified as a regulator of the composition of signaling complexes (*Liang et al., 2013*).

Using a large-scale live cell imaging approach, we were able to distinguish between inefficient cSMAC formation and inefficient recruitment of a signaling intermediate to an existing cSMAC: As central interface accumulation of numerous signaling intermediates was diminished upon costimulation blockade and Itk-deficiency (*Figure 1C,D*) cSMAC formation was most likely impaired. Central interface accumulation of spatially targeted LAT and SLP-76 upon the attenuated T cell stimuli thus has to represent cSMAC restoration as confirmed by CLEM (*Figure 3D*).

Complexes of supramolecular dimensions are part of the cSMAC (*Figure 3B*). In general, supramolecular complex formation becomes more likely with increasing concentrations and valences of the complex components (*Banani et al., 2016*; *Li et al., 2012*). Accordingly, replacement of proline-rich regions in Sos that mediate multivalent LAT/Grb2/Sos interactions leads to reduced LAT clustering and phosphorylation (*Kortum et al., 2013*). Similarly, when LAT phosphorylation (*Figure 2C*), which is required for interactions of LAT with SH2 domain-containing signaling intermediates, was reduced upon costimulation blockade and Itk deficiency LAT clustering at the interface center was also diminished (*Figure 2B*). Compensation for such diminished LAT valence by fusing LAT with additional protein interaction domains restored µm scale central LAT accumulation. The dependence of LAT clustering on the number of functional protein interaction motifs, that is its valence, supports the supramolecular nature of protein complexes within the cSMAC. The importance of valence is further illustrated in the divergent behavior of fusion proteins versus their components. For example, the PKCθ V3 domain in isolation does not localize to the interface center even under full stimulus conditions (*Figure 4—figure supplement 1A*) nor does LAT upon costimulation blockade (*Figure 2B*). However, the LAT-V3 fusion protein shows dominant central localization upon costimulation blockade (*Figure 4B*). How is this possible? By generating a LAT-V3 fusion proteins we didn't only add the localization preferences of LAT and the PKCθ V3 domain, we also increased the valence of the fusion protein over its components as a key facilitator of recruitment to supramolecular complexes (*Li et al., 2012*).

Interestingly, fusion domains could not drive spatial features of adaptor localization by themselves but could only enhance intrinsic adaptor localization preferences. We could enhance central LAT clustering across all time points, central SLP-76 clustering only during the first minute of cell coupling and central Grb2 clustering not at all. This mirrored the accumulation patterns of the non-targeted adaptors under full stimulus conditions (*Figures 4*, *7* and *8*) and therefore strongly suggests that localization of the spatially targeted adaptors was driven by a combination of intrinsic localization motifs and the spatial information provided by the fused domains. In support, while both PKCθ V3 and PLCδ PH did not display central localization in isolation, they could enhance central localization of LAT and SLP-76 (*Figures 4* and *7*). Similarly, artificially enhanced cSMAC formation through expression of LAT V3 did not lead to the recruitment of signaling intermediates with intrinsically weak cSMAC preference such as Vav1 (*Figure 4C*). Our inability to force cSMAC localization implies that even upon addition of new protein interaction domains the overall molecular composition of supramolecular complexes in the cSMAC is fairly well conserved: A core of protein-protein interactions may be required for complex formation such that addition of new protein interactions can only enhance complex stability but not generate complexes of fundamentally different composition. Supramolecular assemblies in the cSMAC thus likely exist in a delicate balance between compositional conservation for complex identity and some redundancy in protein interaction motifs and stoichiometry to allow flexible regulation of stability. As a possible exception to this rule it was only upon the strongest attenuation of T cell activation, Itk-deficiency combined with costimulation

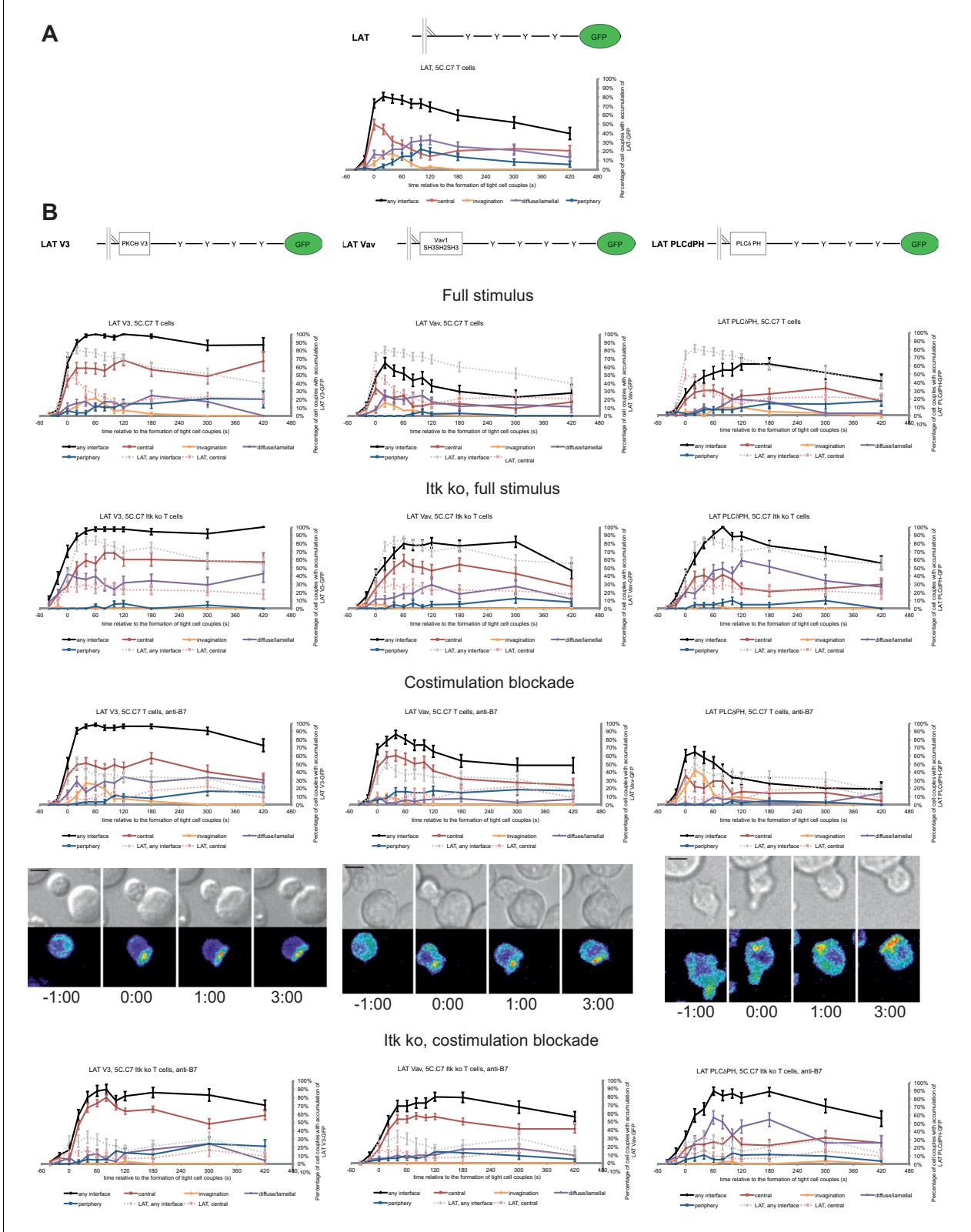

**Figure 4.** LAT localization can be controlled by fusion with protein domains with a strong interface localization preference. (**A**) A schematic representation of LAT-GFP is given (top) with LAT accumulation data under full stimulus conditions (bottom, from *Figure 2B*) as a reference for the rest of the figure. (**B**) On top schematic representations are given for the three fusion proteins of LAT with protein localization domains as indicated. Corresponding imaging data are given in the respective columns below: wild type or Itk-deficient 5C.C7 T cells transduced to express the spatially

*Figure 4 continued on next page*

*Figure 4 continued*

targeted LAT construct indicated on the top of the column were activated with CH27 APCs (10 µM MCC) in the absence or presence of 10 µg/ml anti-CD80 plus anti-CD86 ('costimulation blockade') with different T cell activation conditions given in separate rows as indicated. Each individual graph gives the percentage of cell couples that displayed accumulation of the spatially targeted LAT construct with the indicated patterns as in *Figure 2B* relative to tight cell couple formation in solid colors. Broken gray and red lines indicate accumulation of non-targeted LAT-GFP in any or the central interface pattern, respectively, under the same T cell activation conditions (from *Figure 2B*) for reference. For costimulation blocked conditions representative imaging data are given similar to *Figure 2A*. Corresponding videos are available as *Figure 4—Videos 1–3*. 37–53 cell couples from 2 to 5 independent experiments were analyzed per condition, 551 total. Statistical analysis is given in *Figure 4—source data 1*.

The online version of this article includes the following video, source data, and figure supplement(s) for figure 4:

**Source data 1.** Statistical significance of differences in accumulation of spatially targeted as compared to non-targeted LAT under different T cell activation conditions is given for the indicated patterns as determined by proportion's z-test.
**Figure supplement 1.** Patterning of isolated targeting domains and interface recruitment of LAT-GFP and spatially targeted version thereof.
**Figure 4—video 1.** The video is displayed similar to *Figure 2—Video 1*.
https://elifesciences.org/articles/45789#fig4video1
**Figure 4—video 2.** The video is displayed similar to *Figure 2—Video 1*.
https://elifesciences.org/articles/45789#fig4video2
**Figure 4—video 3.** The video is displayed similar to *Figure 2—Video 1*.
https://elifesciences.org/articles/45789#fig4video3

---

blockade, that Grb2 as one of three signaling intermediates investigated was recruited to a LAT V3-based cMSAC to an extent not seem for Grb2 interface accumulation under other T cell activation conditions (*Figure 6*). We suggest the following scenario. Grb2 doesn't effectively compete for binding to the cSMAC under strong and even moderate T cell activation conditions. Likely, stronger binding signaling intermediates are activated to a sufficient extent under those conditions to outcompete Grb2. However, with the strong signaling attenuation provided by the combination of costimulation blockade and Itk-deficiency activation of more competitive signaling intermediates becomes sufficiently inefficient to allow Grb2 recruitment to the LAT V3-based signaling complexes.

Reconstitution of central LAT and SLP-76 clustering under attenuated T cell activation conditions could restore *IL-2* mRNA generation. The association of lack of Grb2 centrality with lack of an effect on *IL-2* mRNA generation provides a specificity control. Protein clustering in the cSMAC thus was a critical component of efficient T cell signaling. For the cSMAC to enhance T cell function experimental reconstitution of cSMAC formation needed to closely mimic cSMAC features observed with non-targeted constructs under full stimulus conditions. Time-dependent roles of the cSMAC have been proposed before (*Freiberg et al., 2002*) yet are difficult to compare to the work here as the molecules investigated differed. In our work under full stimulus conditions LAT-GFP and SLP-76-GFP were efficiently recruited to the interface center within the first two minutes of T cell activation with diminished recruitment thereafter. Recruitment of LAT and SLP-76 to the interface center upon attenuation of T cell activation by fusion with the Vav1 SH3SH2SH3 and PLCδ PH domains closely reproduced these dynamics (*Figures 4B* and *7*) and largely restored *IL-2* mRNA generation (*Figures 5A* and *7*). However, recruitment of LAT to the interface center to a greater extent and duration by fusion with PKCθ V3 (*Figure 4B*) could not enhance *IL-2* mRNA generation (*Figure 5A*). We therefore suggest that the cSMAC displays two different time-dependent roles. Within the first one to two minutes of T cell activation it efficiently brings together the large number of proximal T cell signaling intermediates required for efficient T cell activation. Subsequently, a substantial number of key signaling intermediates including SLP-76, Itk, PLCγ, and Vav1 leave the cSMAC and move to smaller signaling complexes supported by an interface-wide lamellal actin network (*Roybal et al., 2015*). Retention of a more limited subset of signaling intermediates in the cSMAC after this time thus may render them less accessible to their interaction partners and therefore diminish sustained signal transduction. Signal enhancing and attenuating roles of the cSMAC thus may both occur as regulated by the specific time-dependent composition of the complex.

Itk and costimulation likely contribute to cSMAC formation by overlapping yet partially distinct means. Recruitment to the interface center peaks within the first two minutes of cell coupling for both ligand-engaged CD28 (*Purtic et al., 2005*; *Singleton et al., 2009*) and full length Itk (*Roybal et al., 2015*). Both thus can be expected to provide protein-protein interactions during the early signal amplifying stage of the cSMAC. Itk also has enzymatic activity to directly modify cSMAC

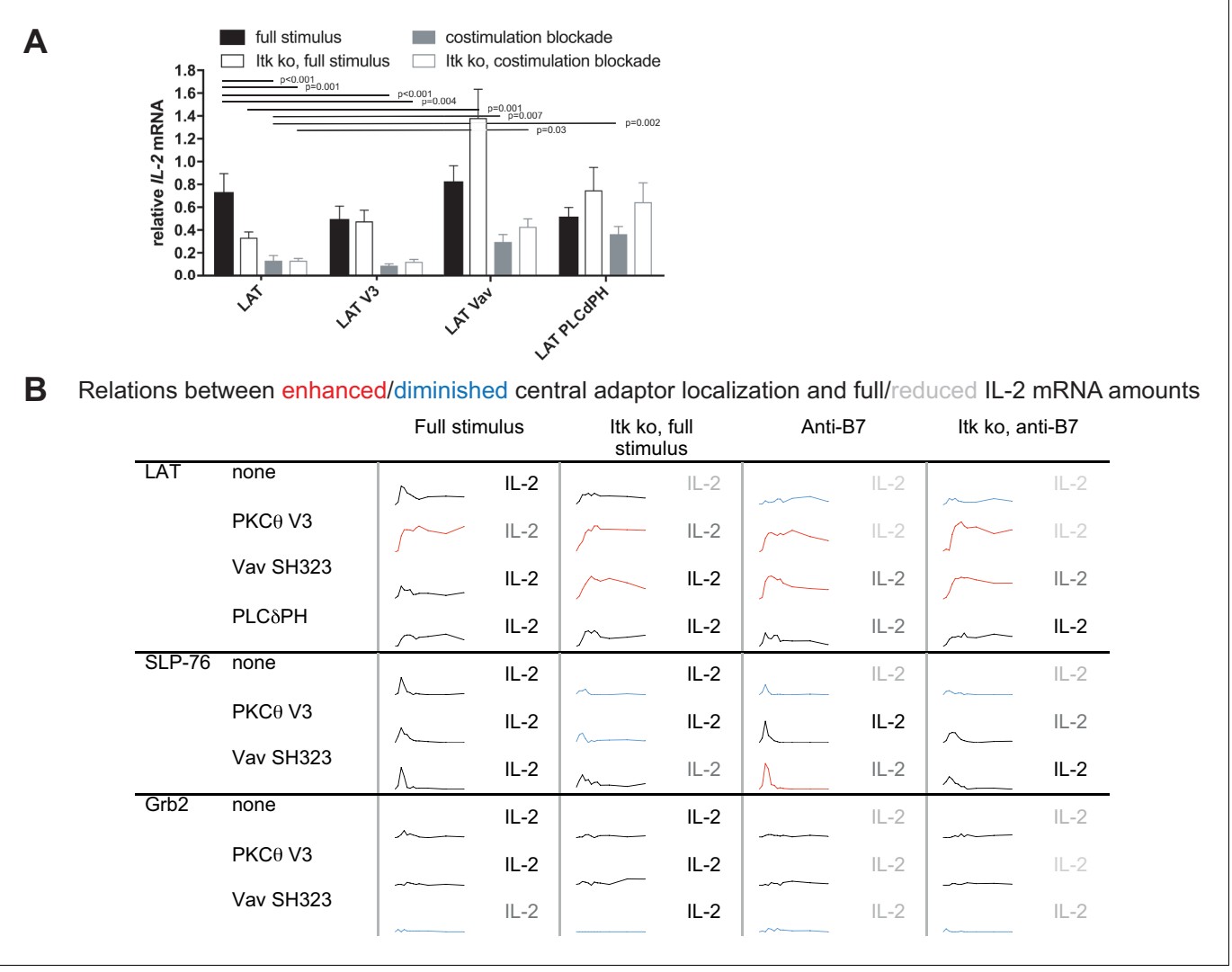

**Figure 5.** Restoration of LAT centrality enhances *IL-2* mRNA generation. (**A**) Wild type or Itk-deficient ('Itk ko') 5C.C7 T cells expressing LAT-GFP or a spatially targeted variant thereof as indicated were activated by CH27 APCs (10 µM MCC) in the absence or presence of 10 µg/ml anti-CD80 plus anti-CD86 ('costimulation blockade'). *IL-2* mRNA amounts are given relative to *IL-2* mRNA in non-transduced 5C.C7 T cells under full stimulus conditions. 3–9 experiments were averaged per condition. Statistical significance of differences as determined by 2-way ANOVA is given for comparisons of interest. (**B**) Sensor accumulation at the interface center and *IL-2* mRNA amounts are summarized. Traces are the percentage cell couples with central accumulation from *Figures 4*, *7* and *8*. Red indicates enhanced central accumulation relative to non-targeted signaling intermediate under full stimulus conditions at at least half of the time points with substantial central accumulation, blue similarly indicates diminished central accumulation. Four shades of gray indicated level of *IL-2* mRNA relative to non-targeted signaling intermediate under full stimulus conditions > 75%, 50–75%, 25–50% and <25%. The online version of this article includes the following figure supplement(s) for figure 5:

**Figure supplement 1.** IL2 mRNA amounts upon expression of the isolated targeting domains.

components. Both costimulation and Itk regulate actin dynamics. However, while costimulation controls core actin turnover through the Arp2/3 complex and Cofilin (*Roybal et al., 2016*), Itk only regulates a SLAT-dependent subset of actin dynamics (*Singleton et al., 2011*). CD28 thus can be expected to contribute more strongly to cSMAC directed transport in complex assembly (*Roybal et al., 2016*)(*Figure 2C*).

In summary, our work establishes that the cSMAC formed in the activation of T cells by APCs consists of multiple supramolecular complexes driven by extensive membrane undulations with a dynamically changing composition. Compositionally rich complexes in the first two minutes of cell

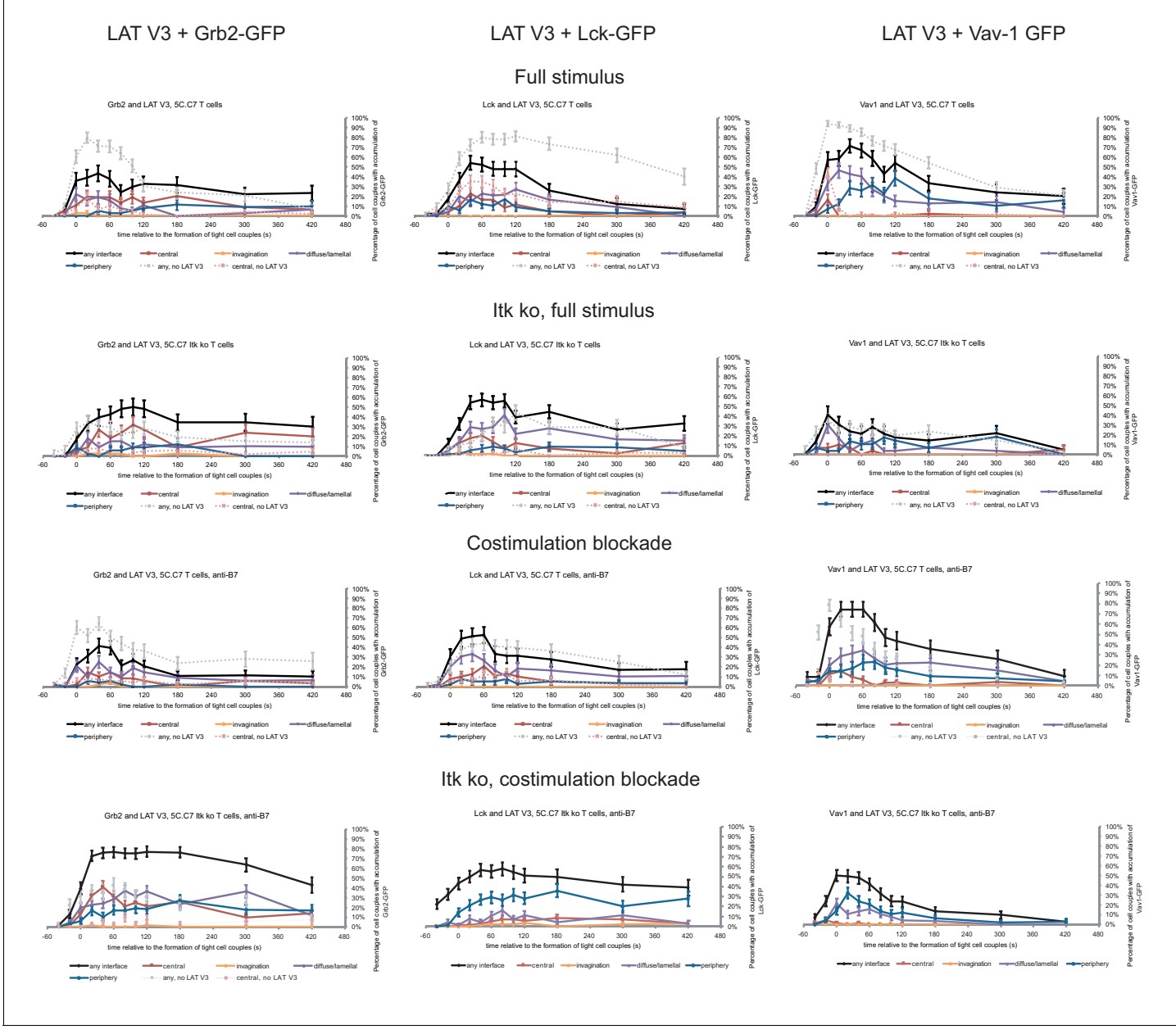

**Figure 6.** Restoration of LAT centrality only modestly affects centrality of other signaling intermediates. Wild type and Itk-deficient 5C.C7 T cells were transduced to express LAT V3 together with the indicated GFP-tagged signaling intermediate and activated with CH27 APCs (10 µM MCC) in the absence or presence of 10 µg/ml anti-CD80 plus anti-CD86 ('costimulation blockade') with different T cell activation conditions given in separate rows as indicated. Each individual graph gives the percentage of cell couples that displayed accumulation of the GFP-tagged signaling intermediate with the indicated patterns as in *Figure 2B* relative to tight cell couple formation in solid colors. Broken gray and red lines indicate accumulation of the signaling intermediate in the absence of LAT V3 in any or the central interface pattern, respectively, under the same T cell activation conditions (from *Figure 8*; *Figure 1—figure supplements 2B–4*). 30–72 cell couples from 2 to 5 independent experiments were analyzed per condition, 582 total. Statistical analysis is given in *Figure 6—source data 1*.

The online version of this article includes the following source data for figure 6:

**Source data 1.** Statistical significance of differences in accumulation of Grb2, Lck and Vav1 in the presence as compared to absence of LATV3 under different T cell activation conditions is given for the indicated patterns as determined by proportion's z-test.

coupling enhanced T cell activation by facilitating efficient signaling interactions whereas thereafter

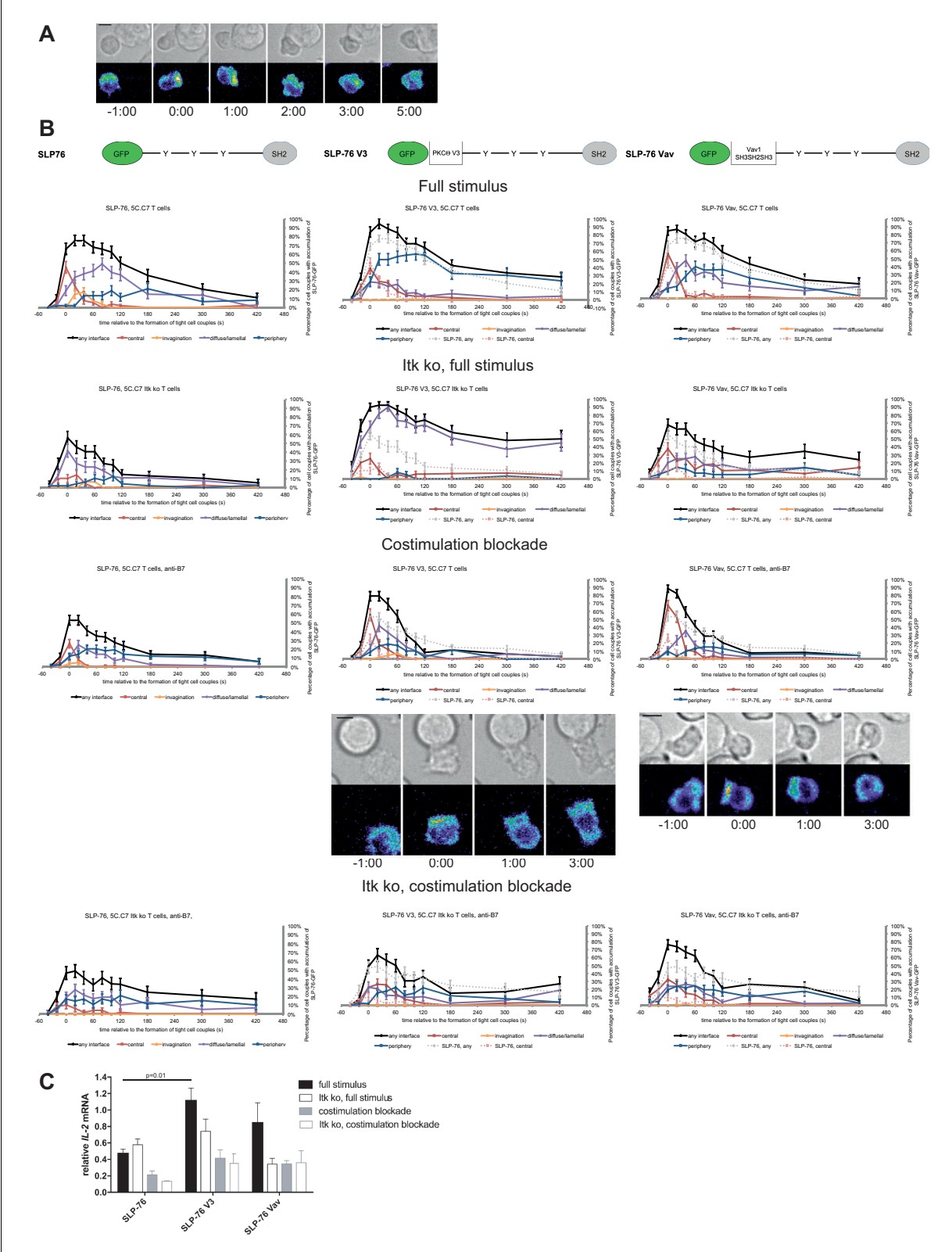

**Figure 7.** SLP-76 localization is regulated by costimulation and Itk and restoration of early SLP-76 centrality enhances *IL-2* mRNA generation. (**A**) An interaction of a SLP-76-GFP-transduced 5C.C7 T cell with a CH27 APC (10 μM MCC) is shown at the indicated time points (in minutes) relative to the time of formation of a tight cell couple as in *Figure 2A*. A corresponding video is available as *Figure 7—Video 1*. (**B**) On top schematic representations are given for SLP-76-GFP and the two fusion proteins of SLP-76 with protein domains as indicated. Corresponding imaging data are

*Figure 7 continued on next page*

*Figure 7 continued*

given in the respective columns below: wild type or Itk-deficient 5C.C7 T cells transduced to express SLP-76 or the spatially targeted SLP-76 construct indicated on the top of the column were activated with CH27 B cell APCs (10 µM MCC) in the absence or presence of 10 µg/ml anti-CD80 plus anti-CD86 ('costimulation blockade') with different T cell activation conditions given in separate rows as indicated. Graphs give the percentage of cell couples that displayed accumulation of the non-targeted (on the left) or spatially targeted SLP-76 construct (middle and right) with the indicated patterns as in *Figure 2B* relative to tight cell couple formation in solid colors. Broken gray and red lines indicate accumulation of non-targeted SLP-76-GFP in any or the central interface pattern, respectively, under the same T cell activation conditions. For costimulation blocked conditions representative imaging data are given below the graphs similar to *Figure 2A*. Corresponding videos are available as *Figure 7—Videos 2* and *3*. 39–83 cell couples from 2 to 5 independent experiments were analyzed per condition, 586 total. Statistical analysis is given in *Figure 7—source data 1*. (C) Wild type or Itk-deficient ('Itk ko') 5C.C7 T cells expressing SLP-76-GFP or a spatially targeted variant thereof as indicated were activated by CH27 APCs (10 µM MCC) in the absence or presence of 10 µg/ml anti-CD80 plus anti-CD86 ('costimulation blockade'). *IL-2* mRNA amounts are given relative to *IL-2* mRNA in non-transduced 5C.C7 T cells under full stimulus conditions. 2–8 experiments were averaged per condition. Statistical significance of differences as determined by 2-way ANOVA is given for a comparison of interest.

The online version of this article includes the following video and source data for figure 7:

**Source data 1.** Statistical significance of differences in SLP-76 accumulation and in accumulation of spatially targeted compared to non-targeted SLP-76 under different T cell activation conditions is given for the indicated patterns as determined by proportion's z-test.

**Figure 7—video 1.** The video is displayed similar to *Figure 2—Video 1*.

https://elifesciences.org/articles/45789#fig7video1

**Figure 7—video 2.** The video is displayed similar to *Figure 2—Video 1*.

https://elifesciences.org/articles/45789#fig7video2

**Figure 7—video 3.** The video is displayed similar to *Figure 2—Video 1*.

https://elifesciences.org/articles/45789#fig7video3

compositionally poorer complexes may sequester signaling intermediates.

# Materials and methods

## Key resources table

| Reagent type (species) or resource | Designation | Source or reference | Identifiers | Additional information |
|---|---|---|---|---|
| Genetic reagent (*M. musculus*) | 5C.C7 TCR transgenic | Davis lab, Stanford (*Seder et al., 1992*) | RRID:MGI:3799371 | |
| Genetic reagent (*M. musculus*) | 5C.C7 TCR transgenic, Itk ko | This paper | | generated by crossing 5C.C7 TCR transgenic with Itk-deficient B6 mice (*Schaeffer et al., 1999*) (RRID:MGI:4356470). |
| Cell line (*Mus musculus*) | CH27 | Davis lab, Stanford | RRID:CVCL_7178 | |
| Cell line (*Homo sapiens*) | Phoenix E | Nolan lab, Stanford | RRID:SCR_003163 | |
| Antibody | Anti - LAT pY191 (Rabbit polyclonal) | Cell Signaling | Cat#3584, RRID:AB_2157728 | WB (1:1000), Immunostaining (1:100) |
| Antibody | Anti-LAT (Rabbit polyclonal) | Cell Signaling | Cat#9166, RRID:AB_2283298 | WB (1:1000), Immunostaining (1:50) |
| Antibody | Anti-rabbit IgG, Alexa Fluor 488 (Donkey polyclonal) | Molecular Probes | Cat#R37118, 1:1000, RRID:AB_2556546 | Immunostaining (1:1000) |
| Antibody | Anti - GAPDH Clone 14C10 (Rabbit monoclonal) | Cell Signaling | Cat#2118, RRID:AB_561053 | WB (1:1000) |

*Continued on next page*

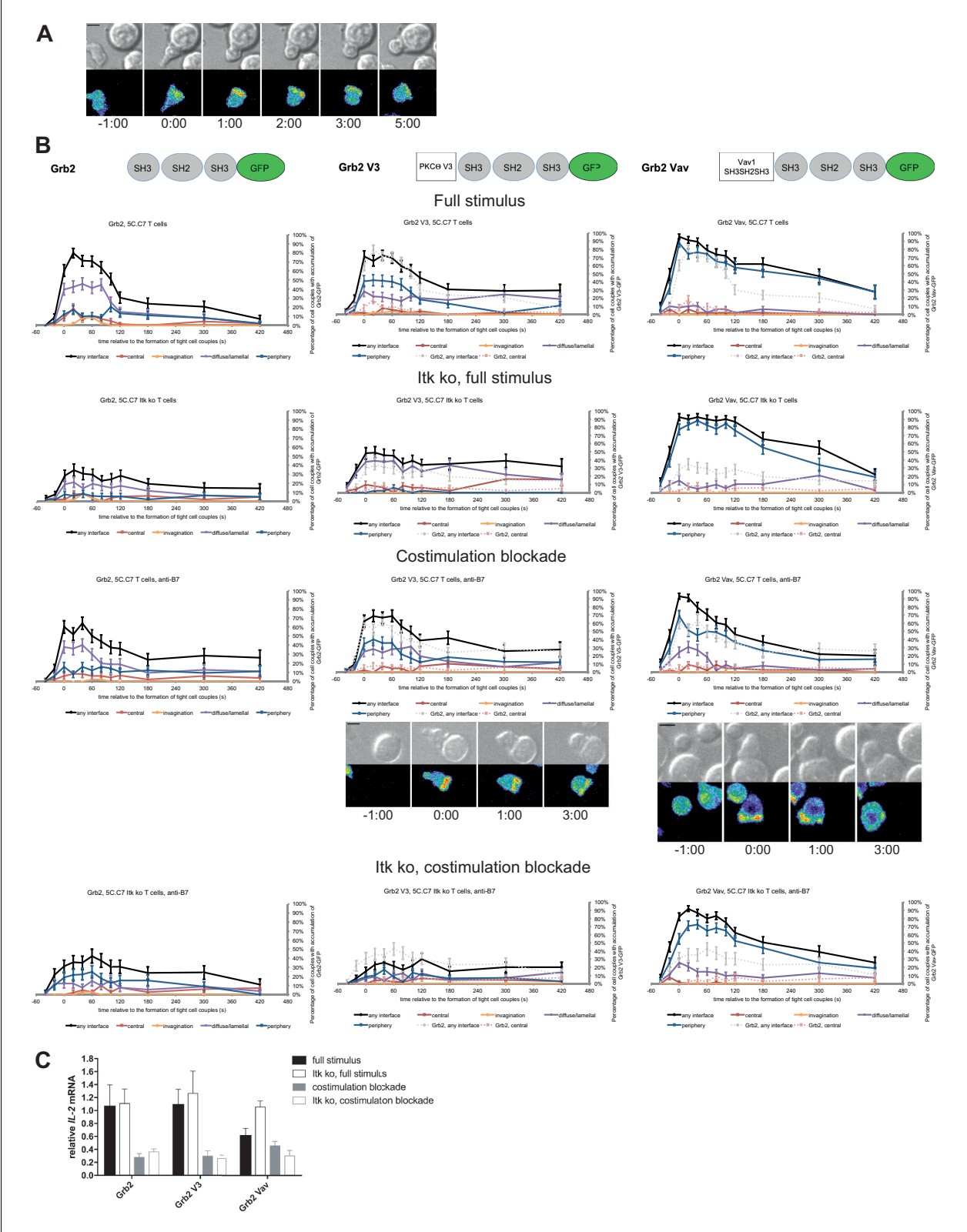

**Figure 8.** Grb2 localization is regulated by costimulation and Itk and doesn't regulate *IL-2* mRNA generation. (**A**) An interaction of a Grb2-GFP-transduced 5C.C7 T cell with a CH27 APC (10 μM MCC) is shown at the indicated time points (in minutes) relative to the time of formation of a tight cell couple as in *Figure 2A*. A corresponding video is available as *Figure 8—Video 1*. (**B**) On top schematic representations are given for Grb2-GFP and the two fusion proteins of Grb2 with protein domains as indicated. Corresponding imaging data are given in the respective columns below: wild type

*Figure 8 continued on next page*

*Figure 8 continued*

or Itk-deficient 5C.C7 T cells transduced to express the spatially targeted Grb2 construct indicated on the top of the column were activated with CH27 APCs (10 µM MCC) in the absence or presence of 10 µg/ml anti-CD80 plus anti-CD86 ('costimulation blockade') with different T cell activation conditions given in separate rows as indicated. Graphs give the percentage of cell couples that displayed accumulation of the non-targeted (on the left for reference) or spatially targeted Grb2 construct (middle and right) with the indicated patterns as in *Figure 2B* relative to tight cell couple formation in solid colors. Broken gray and red lines indicate accumulation of non-targeted Grb2-GFP in any or the central interface pattern, respectively, under the same T cell activation conditions. For costimulation blocked conditions representative imaging data are given below the graphs similar to *Figure 2A*. Corresponding videos are available as *Figure 8—Videos 2* and *3*. 41–62 cell couples from 2 to 5 independent experiments were analyzed per condition, 591 total. Statistical analysis is given in *Figure 8—source data 1*. (C) Wild type or Itk-deficient ('Itk ko') 5C.C7 T cells expressing Grb2-GFP or a spatially targeted variant thereof as indicated were activated by CH27 APCs (10 µM MCC) in the absence or presence of 10 µg/ml anti-CD80 plus anti-CD86 ('costimulation blockade'). *IL-2* mRNA amounts are given relative to *IL-2* mRNA in non-transduced 5C.C7 T cells under full stimulus conditions. 2–5 experiments were averaged per condition.

The online version of this article includes the following video and source data for figure 8:

**Source data 1.** Statistical significance of differences in Grb2 accumulation and in accumulation of spatially targeted as compared to non-targeted Grb2 under different T cell activation conditions is given for the indicated patterns as determined by proportion's z-test.

**Figure 8—video 1.** The video is displayed similar to *Figure 2—Video 1*.

https://elifesciences.org/articles/45789#fig8video1

**Figure 8—video 2.** The video is displayed similar to *Figure 2—Video 1*.

https://elifesciences.org/articles/45789#fig8video2

**Figure 8—video 3.** The video is displayed similar to *Figure 2—Video 1*.

https://elifesciences.org/articles/45789#fig8video3

*Continued*

| Reagent type (species) or resource | Designation | Source or reference | Identifiers | Additional information |
|---|---|---|---|---|
| Antibody | Anti- alpha Tubulin Clone DM1A (Mouse monoclonal) | Thermo Fisher Scientific | Cat#62204, RRID:AB_1965960 | WB (1:1000) |
| Antibody | Anti – CD80 Clone 16–10-A1 (Armenian Hamster monoclonal) | BD Pharmingen | Cat#553736 | Blocking at 10 µg/ml |
| Antibody | Anti – CD86 Clone GL1 (Rat monoclonal) | BD Pharmingen | Cat#553689 | Blocking at 10 µg/ml |
| Peptide, recombinant protein | MCC | Davis lab, Stanford | | Sequence: ANERADLIAYLKQATK |
| Commercial assay or kit | RNeasy Micro Kit | Qiagen | Cat#74004 | |
| Commercial assay or kit | AMV First-Strand cDNA synthesis kit | Invitrogen | Cat#12328032 | |
| Commercial assay or kit | SYBR Green PCR master mix | Life Technologies | Cat#4344463 | |
| Commercial assay or kit | IL-2 OptEIA ELISA | BD Biosciences | Cat#555148 | |
| Sequence-based reagent | qPCR Oligonucleotides, IL-2 | This paper | | AGCTGTTGATGGACCTA and CGCAGAGGT CCAAGTTCAT |
| Software, algorithm | Metamorph image analysis software | Molecular Devices | RRID:SCR_002368 | |
| Software, algorithm | Amira image analysis software | VSG | RRID:SCR_007353 | |

## Antibodies and reagents

Antibody used for quantitative western blotting were α-LAT pY191 (Cell Signaling, #3584, RRID:AB_2157728), α-GAPDH Clone 14C10 (Cell Signaling, #2118, RRID:AB_561053), and α-alpha Tubulin

Clone DM1A (ThermoFisher Scientific, #62204, RRID:AB_1965960). Antibodies used for the blockade of CD80- and CD86-dependent CD28 costimulation were anti-mouse CD80 Clone 16–10-A1 (BD Pharmingen #553736) and anti-CD86 Clone GL1 (BD Pharmingen #553689). Protein transduction versions of constitutively active cofilin (S3A) and Rac1 (Q61L) were purified from *E. coli* and introduced into primary 5C.C7 T cells by 30 min incubation as previously described (*Roybal et al., 2016*).

## Mice and cells

Itk-deficient 5C.C7 mice were generated by crossing B10.BR 5C.C7 TCR transgenic mice (*Seder et al., 1992*)(RRID:MGI:3799371) with Itk-deficient B6 mice (*Schaeffer et al., 1999*)(RRID: MGI:4356470). T cells expanded from the lymph nodes of wild type or Itk-deficient 5C.C7 TCR transgenic mice were used for all experiments. The 5C.C7 TCR recognizes the moth cytochrome c peptide fragment (amino acid residues 88 to 103, ANERADLIAYLKQATK) in the context of I-E$^k$. Single-cell suspensions were made from the lymph nodes of 6- to 8-week-old mice of either gender. The cells were adjusted to $4 \times 10^6$ cells/ml and MCC peptide was added to a final concentration of 3 µM. T cells were transduced with MMLV-derived retroviruses for the expression of signaling sensors, commonly signaling intermediates fused with GFP as described in detail (*Ambler et al., 2017*; *Roybal et al., 2015*; *Singleton et al., 2009*). All animals were maintained in pathogen-free animal facilities at the University of Bristol under the University mouse breeding Home Office License P10DC2972. The CH27 B cell lymphoma cell line (RRID:CVCL_7178) was used in all experiments as APCs. It was shown to be mycoplasm-free using the ATCC Universal Mycoplasm Testing Kit and verified by staining for I-E$^k$, CD80, CD86 and CD54 (ICAM-1). To peptide load the APCs, the CH27 cells were incubated in the presence of 10 µM MCC peptide for at least 4 hr. All cells were maintained in medium composed of RPMI with L-glutamine, 10% fetal bovine serum (FBS, Hyclone), penicillin (100 IU/mL), streptomycin (100 µg/ml), and 0.5 µM β-mercaptoethanol. Interleukin-2 (IL-2)(TECIN recombinant human IL-2, NCI Biological Resource Branch) was added at a final concentration of 0.05 U/ml during parts of the retroviral transduction procedure.

## Time-lapsed imaging of T cell:APC interactions

Our imaging and image analysis protocols have recently been described in great detail in a dedicated publication (*Ambler et al., 2017*). Briefly, time-lapse fluorescence microscopy was performed with retrovirally transduced T cells, FACS-sorted to the lowest detectable sensor expression of 2 µM, and CH27 cells loaded with 10 µM MCC. The T cells and CH27 cells were imaged in imaging buffer (PBS, 10% FBS, 1 mM CaCl$_2$, 0.5 mM MgCl$_2$) on 384-well glass-bottom plates. All imaging was performed on a Perkin Elmer UltraVIEW ERS 6FE spinning disk confocal systems fitted onto a Leica DM I6000 microscope body equipped with full environmental control and a Hamamatsu C9100-50 EMCCD. A Leica 40x HCX PL APO oil objective (NA = 1.25) was used for all imaging. Automated control of the microscope was performed with Volocity software (Perkin Elmer). For experiments in which the CD80- and CD86-dependent activation of CD28 was blocked, peptide-loaded CH27 cells were incubated on ice for 30 min in the presence of anti- CD80 Clone 16–10-A1 (10 µg/ml) and anti- CD86 Clone GL1 (10 µg/ml)(BD Pharmingen) antibody before the CH27 cells were transferred to the imaging plate with the T cells. For experiments in which cells were reconstituted with protein transduction versions of constitutively active Rac and cofilin, T cells were incubated for 30 min at 37°C with the protein transduction reagents at the indicated concentrations in the imaging plate before the addition of the peptide-loaded CH27 cells. Each time-lapse image sequence was generated by taking a differential interference contrast brightfield image and a 3D image stack of the GFP channel every 20 s for 46 frames at 37°C. Voxels in these 3D images were of size 0.34 µm in the horizontal plane and 1 µm along the optical axis. For long-term NFκB-GFP imaging, the glass bottom of the imaging plate was coated with 10 µg/ml anti-CD CD19 antibody (BD Pharmingen #553784) to prevent APC from moving and a Pathological Devices LiveCell chamber was fitted over the imaging plate to prevent evaporation. Nuclear localization of NFκB-GFP was determined as in *Roybal et al. (2015)*.

## Image analysis

The location and frame number of each T cell:APC couple were identified as when either the T cell: APC interface had reached its full width or the cells had been in contact for 40 s, whichever came

first. Patterns of signaling sensor enrichment were assessed according to previously established quantitative criteria (Figure 2 in *Singleton et al., 2009*) as depicted in the *Figure 1—figure supplement 2A*. Briefly, the six, mutually exclusive interface patterns were: accumulation at the center of the T cell-APC interface (central), accumulation in a large T cell invagination (invagination), accumulation that covered the cell cortex across central and peripheral regions (diffuse), accumulation in a broad interface lamella (lamellal), accumulation at the periphery of the interface (peripheral) or in smaller protrusions (asymmetric). Briefly, for each cell couple at each time point we first determined whether fluorescence intensity in the area of accumulation was >40% above the cellular fluorescence background. If so, the geometrical features of the area of accumulation, fraction of the interface covered, location within the interface, and extension of the area of accumulation away from the interface (Figure 2 in *Singleton et al., 2009*), were used to assign the cell couple to one of the mutually exclusive patterns. Systems-scale cluster analysis was performed with Cluster (Michael Eisen, UC Berkeley) as established (*Singleton et al., 2009*).

To measure interface enrichment of LAT and spatially targeted versions thereof we used a recently developed computational image analysis routine (*Roybal et al., 2016*). Very briefly, starting with the manual cell couple identification described above T cells were segmented, reoriented with the T cell:APC interface facing up using the 'two-point synapse annotation' procedure (*Roybal et al., 2016*), and the cell shape was standardized to a half spheroid to allow voxel-by-voxel comparison across all cell couples analyzed. After transformation to the standard shape, the fluorescence distribution in each cell at a given time point was represented as a standardized vector (of length 6628) formed from the intensity values of each of the voxels within the template shape, where the intensities for each time point were normalized so that the values of the vector were probabilities (that is, fractions of total intensity). To measure interface enrichment, we defined an interface enrichment region as the 10% most fluorescent voxels of the average probability distribution across all cells, for all time points, and for all sensors. We defined enrichment to be the ratio of the mean probability in the distribution of that sensor for that cell at that time point within the interface enrichment region and the mean probability in the entire cell.

## STED imaging

CH27 APCs were adhered to $\alpha$-CD19 antibody-coated coverslips. T cells were then allowed to interact with APCs for 4.5 min and fixed with 4% PFA for 20 min at 4°C followed by PFA quenching using ammonium chloride for 10 min at 4°C. Cells were permeabilized for 20 min in 0.02% Triton X-100 in PBS at 4°C. T cells were blocked in 1% BSA in PBS for 30 min at room temperature and probed with primary antibodies against LAT (1:100, Cell Signalling #9166, RRID:AB_2283298) or phospho-LAT Tyr 191 (1:50, Cell Signalling #3584, RRID:AB_2157728) diluted in 1% BSA with Fc block (Rat Anti-Mouse CD16/CD32, #553141, BD Bioscience, RRID:AB_394656) at the same dilution in PBS for overnight at 4°C. Cells were washed with PBS three times and incubated with secondary antibody, Donkey anti-rabbit IgG, Alexa Fluor 488 (Molecular Probes #R37118, 1:1000, RRID:AB_2556546) in 1% BSA with Fc block (1:500) for 1 hr at room temperature. Coverslips were washed with PBS before mounting using ProLong Gold (Thermo Fisher) and cured for 24 hr at room temperature.

Fixed cells were imaged through a 100x HC PL APO CS2 1.4 NA objective on a Leica SP8 AOBS confocal laser scanning microscope. Alexa Fluor 488 was excited using a white light laser with an emission filter between 498–520 nm and STED depletion was achieved using a 592 nm continuous wave fibre laser. Images were first de-convolved using Hyugen Professional followed by automated puncta analysis with ImageJ (NIH). LAT puncta were detected and measured using Wolfson Bioimaging ImageJ plugins (Modular Image Analysis). Individual puncta were identified using the Otsu algorithm (*Otsu, 1979*) with a threshold multiplier of 3.5 A.U. followed by a filtration mode of the Watershed 3D method to identify separate puncta. Small puncta detected in the APCs that don't express LAT were used to derive a detection size threshold for LAT complexes such that all puncta smaller than the 95 percentile of the APC puncta size distribution were excluded from the analysis. Thus, complexes smaller than 0.04/0.06 $\mu m^3$ in the LAT/pLAT data were excluded from the analysis. Repeating the analysis with a 99-percentile cut-off didn't change the conclusions reached. The closest distance between adjacent puncta detected was 100 nm.

## CLEM

The interaction of 5C.C7 T cells with CH27 B cell APCs was imaged live by spinning disk confocal microscopy as described above using a 35 mm glass bottom finder dish (Mattek). Upon formation of a cSMAC, cells were immediately fixed using 2.5% Glutaraldehyde (Agar Scientific) in 0.1M cacodylate buffer, stained in 1% osmium tetroxide (EMS) in cacodylate buffer, dehydrated and embedded in Epon812 resin (TAAB) for 24 hr at 60°C. Samples were removed from EPON and trimmed to section of interest. Trimmed samples were sectioned at 300 nm using an Ultramicrotome (Leica, EM UC7) with a diamond knife (DiATOME) and stained with uranyl acetate and lead citrate (Agar Scientific). Sections were analyzed using a FEI Tecnai 12 120kV BioTwin equipped with a bottom-mount 4*4K Eagle CCD camera. The tomogram data series was acquired using a FRI Tecnai 20 TEM between −50° to +50° with a 2.0° increment. The data were reconstructed using IMOD etomo software. Segmentation was made using AMIRA software (VSG), while for visualization, a combination of IMOD, AMIRA and Image J were used. To determine membrane undulations in the cSMAC region, the interface diameter was divided into four equal sections with the central two sections defined as the cSMAC, a conservative assumption as cSMACs commonly were smaller than half of the interface diameter.

## IL-2 ELISA

Live wild type or Itk-deficient 5C.C7 T cells were FACS sorted to generate comparable cell numbers across each assay. CH27 B cells were peptide loaded with 10 µM MCC for four hours or overnight. T and B cells were mixed in round bottom 96-well plates at $1 \times 10^4$ T cells to $5 \times 10^4$ B cells. For costimulation blockade 10 µg/ml α-CD80 and α-CD86 were added to each well. The cells were then incubated for 18 hr and IL-2 amounts in the supernatant were determined using a mouse IL-2 OptEIA ELISA kit from BD Biosciences as per manufacturer's instructions.

## *IL-2* mRNA determination

CH27 B cell lymphoma APCs were peptide loaded overnight with 10 µM MCC peptide. Live wild type or Itk-deficient 5C.C7 T cells, non-transduced or expressing adaptor protein-GFP or targeted variants thereof, were FACS sorted to generate comparable cell numbers across each assay. $1 \times 10^4$ T cells and $5 \times 10^4$ APCs cells were centrifuged for 30 s at 1,000 rpm to maximize cell-to-cell contact and incubated at 37°C for 2 hr. For costimulation blockade 10 µg/ml α-CD80 and α-CD86 were added to each well. mRNA was isolated using the Qiagen RNeasy Micro Kit (Qiagen, UK) according to manufacturer's instructions. cDNA was generated using an Invitrogen AMV First-Strand cDNA synthesis kit (Life Technologies, UK) according to manufacturer's instructions. *IL-2* mRNA amounts were determined with a SYBR Green PCR master mix from Life Technologies (4344463) relative to mRNA for β−2 microglobulin on a DNA Engine Opticon II System (Bio-Rad) using the following oligonucleotides, *IL-2*: AGCTGTTGATGGACCTA and CGC AGA GGT CCA AGT TCA T, *β−2 microglobulin*: GCTATCCAGAAAACCCCTCAA and CGG GTG GAA CTG TGT GTT ACG T.

## Western blotting analysis and GFP pulldowns

CH27 B cell lymphoma APCs were peptide loaded overnight with 10 µM MCC peptide. Live wild type or Itk-deficient 5C.C7 T cells, non-transduced or expressing LAT-GFP or a targeted variant thereof, were FACS sorted to generate comparable cell numbers across each assay. $1 \times 10^6$ T cells and $1 \times 10^6$ APCs cells were centrifuged for 30 s at 1,000 rpm to maximize cell-to-cell contact and incubated at 37°C for the indicated time. Subsequently, samples were immediately lysed with cold RIPA lysis buffer (Millipore) plus protease/phosphatase inhibitor cocktail (Cell Signaling) for 30 min on ice. To remove the insoluble fraction, samples were centrifuged at 20,000 g for 15 min. Supernatant were run on SDS/PAGE gels, transferred to PDVF membranes and blotted according to standard protocols. Blots were stripped and reprobed with an anti-GAPDH or anti-α tubulin antibody to normalize for sample loading.

The addition of protein interaction domains to LAT is predicted to alter their interactome. To identify proteins that selectively interact with the interaction domain-LAT fusion proteins, we used the GFP tag of the fusion proteins in a pull-down experiment. As spatiotemporal organization and expression of key signaling intermediates, prominently phosphatase and tensin homolg (PTEN), commonly differ between primary and immortalized T cells, we performed these experiments using

primary murine T cells. LAT-interaction domain-GFP fusion proteins (*Figure 4B*) were expressed in primary T cells by retroviral transduction. As transduction efficiencies of 5C.C7 T cells were not sufficiently high to generate at least $5x10^6$ transduced T cells, we used $CD8^+$ CL4 TCR transgenic T cells (*Jenkinson et al., 2005*). To maximize protein interactions, CL4 T cells were activated with pervanadate (0.1mM vanadate plus 3mM hydrogen peroxide). T cells were lysed in RIPA buffer and GFP fusion proteins were pulled down with GFP-trap beads (Chromotek, #GTA20) according to manufacturer's instruction. Bead eluates were analyzed on Coomassie-stained gels and specific bands could not be identified on top of a continuous signal. There are possible experimental constraints. T cells are small with 70% of the cell volume comprised by the nucleus (*Roybal et al., 2015*). While we could FACS sort up to $5x10^6$ retrovirally transduced T cells, a strong pull-down signal would have likely required five times as many cells. Protein interactions triggered by pervanadate may less specific than those in the cSMAC. Protein accumulation in the cSMAC enriches less than 10% of the total cellular amount of an accumulated protein (*Singleton et al., 2009*). While such enrichment is readily detectable by imaging, biochemical detection on the background of the larger non-accumulated cellular pool of a protein may be more demanding.

## Statistics

The frequency of occurrence of interface accumulation patterns was analyzed pairwise with a proportion's z-test as reported in figure supplements. Data from independent experiments were pooled after establishing that they were not significantly different. p values were not corrected for multiple comparisons as the corresponding pFDR q-values (*Storey et al., 2004*) were similar. Images of 50–100 cell couples are acquired per condition. A difference in pattern occurrence of 30% between two experimental conditions can be detected with a power of 0.8. *IL-2* mRNA amounts were first logarithmically transformed to stabilize the variance and approximate to the normal distribution. Outliers were identified using Chauvenet's criterion. Resulting data were analyzed by 1-way ANOVA with Tukey's adjustment for multiple comparisons or 2-way ANOVA with the Sidak adjustment for multiple comparisons depending on the number of variables compared. IL-2 ELISA data were first logarithmically transformed and then analyzed by 1-way ANOVA with Tukey's adjustment for multiple comparisons. LAT phosphorylation data were first logarithmically transformed and then analyzed by 1-way ANOVA with Tukey's adjustment for multiple comparisons separately for each time point.

## Acknowledgements

We acknowledge the University of Bristol FACS and Wolfson BioImaging facilities for providing equipment and technical support and Kole Roybal for the execution of the long-term NFκB imaging experiments. The work was supported by grants from the US National Science Foundation (MCB1121793 to CW, MCB1121919 and MCB1616492 to RFM), ERC (CW: PCIG11-GA-2012–321554), BBSRC (CW: BB/P011578/1) and US National Institutes of Health (P41 GM103712).

## Additional information

### Funding

| Funder | Grant reference number | Author |
| --- | --- | --- |
| Biotechnology and Biological Sciences Research Council | BB/P011578/1 | Laura E McMillan<br>Sin Lih Tan<br>Gaia Bellomo<br>Clementine Massoue<br>Christoph Wülfing |
| H2020 European Research Council | PCIG11-GA-2012-321554 | Danielle J Clark<br>Christoph Wülfing |
| National Science Foundation | MCB1121793 | Minna Du<br>Christoph Wülfing |
| National Science Foundation | MCB1616492 | Xiongtao Ruan<br>Robert F Murphy |
| National Institutes of Health | P41 GM103712 | Xiongtao Ruan |

| | | Robert F Murphy |
| --- | --- | --- |
| National Science Foundation | MCB1121919 | Robert F Murphy<br>Xiongtao Ruan |

The funders had no role in study design, data collection and interpretation, or the decision to submit the work for publication.

## Author contributions

Danielle J Clark, Laura E McMillan, Sin Lih Tan, Gaia Bellomo, Clementine Massoue, Kentner L Singleton, Minna Du, Formal analysis, Investigation; Harry Thompson, Lidiya Mykhaylechko, Investigation; Dominic Alibhai, Xiongtao Ruan, Alan Hedges, Formal analysis; Pamela L Schwartzberg, Resources; Paul Verkade, Conceptualization, Formal analysis, Investigation; Robert F Murphy, Formal analysis, Supervision, Funding acquisition; Christoph Wülfing, Conceptualization, Data curation, Formal analysis, Supervision, Funding acquisition, Investigation, Writing—original draft, Project administration, Writing—review and editing

## Author ORCIDs

Robert F Murphy (iD) https://orcid.org/0000-0003-0358-901X
Christoph Wülfing (iD) https://orcid.org/0000-0002-6156-9861

## Ethics

Animal experimentation: All animals were maintained in pathogen-free animal facilities at the University of Bristol under a University mouse breeding Home Office License.

## Decision letter and Author response

Decision letter https://doi.org/10.7554/eLife.45789.sa1
Author response https://doi.org/10.7554/eLife.45789.sa2

# Additional files

## Supplementary files

• Transparent reporting form

## Data availability

All imaging data are openly accessible via figshare (http://doi.org/10.1184/R1/9963566) and LAT phosphorylation data that support the findings of this study are available at the University of Bristol data repository (https://doi.org/10.5523/bris.2uoex1k196c4o2c80eddeekf04).

The following datasets were generated:

| Author(s) | Year | Dataset title | Dataset URL | Database and Identifier |
| --- | --- | --- | --- | --- |
| Danielle J Clark, Laura E McMillan, Sin Lih Tan, Gaia Bellomo, Clémentine Massoué, Harry Thompson, Lidiya Mykhaylechko, Dominic Alibhai, Xiongtao Ruan, Kentner L Singleton, Minna Du, Alan J Hedges, Pamela L Schwartzberg, Paul Verkade, Robert F Murphy, Christoph Wülfing | 2019 | Data in support of Clark et al. | https://doi.org/10.5523/bris.2uoex1k196c4o2-c80eddeekf04 | University of Bristol data repository, 10.5523/bris.2uoex1k196c4o2c80eddeekf04 |
| Danielle J Clark, | 2019 | Image data from Transient protein | http://doi.org/10.1184/ | figshare, 10.1184/R1/ |

Laura E McMillan, Sin Lih Tan, Gaia Bellomo, Clementine Massoue, Harry Thompson, Lidiya Mykhaylechko, Dominic Alibhai, Xiongtao Ruan, Kentner L Singleton, Minna Du, Alan Hedges, Pamela L Schwartzberg, Paul Verkade, Robert F Murphy, Christoph Wülfing

accumulation at thecenter of the T cell antigen-presenting cellinterface drives efficient IL-2 secretion

R1/9963566

9963566

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
