## [Decision Letter]

Thank you for submitting your article "Transient protein accumulation at the center of the T cell antigen presenting cell interface drives IL-2 secretion" for consideration by *eLife*. Your article has been reviewed by three peer reviewers, including Michael L Dustin as the Reviewing Editor and Reviewer #1, and the evaluation has been overseen by Arup Chakraborty as the Senior Editor.

The reviewers have discussed the reviews with one another and the Reviewing Editor has drafted this decision to help you prepare a revised submission.

Summary:

Your work is commended for quantitative analysis of supramolecular structures with a focus on the SMACs including the use of chimeric signaling adapter to better understand the role of costimulation and the Itk kinase in supramolecular structures and function. However, there were a number of concerns raised. While there are many issues, most can be addressed without new data, but relate significantly to the organisation of information. The comments where new data appear to be necessary to address the concern are marked with asterisks (*).

1) Regarding inconsistency in IL-2 protein and message:

a) The data presented in Figure 1A and B are inconsistent. On the one hand, it is demonstrated that IL-2 amounts secreted to the supernatant of full stimulus T cells vs. full stimulus Itk KO at 10 µM is relatively unchanged, but strangely, on the other hand, the IL-2 mRNA is drastically lower in the Itk KO. No explanations are provided for this inconsistency between the protein and the mRNA levels. This issue should be addressed by the authors.

* Could you provide a time course of IL-2 accumulation in comparison of IL-2 mRNA to provide a clear explanation and bolster the physiological significant of the subsequent investigation.

b) Figure 1C and D. The authors profile the signaling molecules at the different compartments of the IS at different settings, i.e., full stimulation in comparison with co-stimulation blockade (C) and full stimulation vs. full stimulus Itk deficiency (D). However, it is difficult to deduce clear-cut conclusions since the profile of signaling molecules in C is much wider than in D. The authors should explain why they neglect specific molecules in D that were analyzed in C. In addition, the authors should include the profile of signaling molecule localization for the combined treatment of co-stimulation blockade and Itk knockout, as these appear to create an additive effect. Similarly, for Figure 1—figure supplement 2B, Figure 1—figure supplement 3 and Figure 1—figure supplement 4, the graphs of full stimulation vs. co-stimulation blockade/Itk deficiency should compare similar signaling molecular kinetics. Also, Figure 1—figure supplement 3 shows the kinetics of Akt (co-stimulation blockade) that is missing in Figures 1—figure supplements 2 and 4 (Itk deficiency), and Figure 1—figure supplement 4 shows the kinetics of PKC that is missing in Figure 1—figure supplements 2 and 3.

c) In Figure 5, the authors should show IL-2 concentration in addition to IL-2 mRNA amounts due to the inconsistency between them in Figure 1. In addition, the authors show that LAT-V3 restored LAT centralization under co-stimulation inhibition/Itk deficiency, and that LAT-V3 partially restores Grb-2 and Lck centralization under Itk deficiency, yet it had no effect on IL-2 mRNA. This should be discussed by the authors, as it suggests that it is more than LAT centralization that affects IL-2 production.

d) In Figure 5, the authors showed Grb-2, Lck, and VAV-1 centralization as a function of LAT-V3 presence upon stimulation, but omitted the LAT-VAV and LAT- PLCδPH constructs and how they affect signaling molecule centralization.

* These should be included in the data as they also rescued LAT centralization. In addition, the authors should show how the different LAT constructs under the combined treatment of Itk and the co-stimulation blockade influence Grb-2, Lck, and VAV-1, as this may have an additive effect that can be noticeable.

2) Missing information: a) * In the WBs of LAT phosphorylation (Figure 2—figure supplement 2 and Figure 4—figure supplement 1F) the unstimulated time point (often called 0 min stimulation) is missing. Thus, the fold increase upon stimulation remains unknown. Similarly, the total amount of LAT needs to be shown; e.g. in S5D GFP-LATV3 is less phosphorylated than wt GFP-LAT. Was this due to lower expression levels of LATV3 and per molecule there was actually more LATV3 phosphorylation than wt LAT phosphorylation?

b) * Relative expression levels of wt to mutant molecules need to be shown for the most important experiments.

c) It’s not exactly clear how the authors go from the STED results to the volumes. In order to be able to do this the authors would need to report the resolution of the STED system. Is resolution with STED will have a tradeoff between lateral and axial resolution the best isotropic resolution for 3D step might be on the order of 80 nm xy and 100 nm z. Higher xy resolution might be achieved, but only at cost of degraded z resolution.

d) The correlative fluorescence-EM seems to be focused on temporal correlation but it’s not clear if they are look at the same call imaged by fluorescence and if they can register the fluorescence and EM data? Can they really say what structure is the fluorescence reported cSMAC in a given EM view. * A supplementary figure could be constructed to better illustrate how and to what level they are look at correlations.

3) Chimeric receptors:

a) Figure 3. The authors should introduce the fused V3 domain to LAT when it is first used, and present the goal of this experiment. I assume that the authors used LAT-V3 under co-stimulation blockade to demonstrate that it rescue membrane undulation; however, this is not mentioned. Did co-stimulation blockade abrogate membrane undulation in Figure 3? Figure 3D does not show these data. Although LAT-V3 is mentioned in Figure 3, it is explained only in Figure 4.

b) * Equipping LAT (but also SLP76 and Grb2) with new interaction domains is supposed to generate chimeric molecules that bind to new binding partners. For example, LATVav should also bind to proteins that normally would bind to Vav. This should be tested by classical immunoprecipitation experiments (and if this is difficult due to low cell numbers in the primary cells, one could do it with stable cell lines expressing the chimeric molecules). Without these experiments the new interactions remain hypothetical.

c) The chimeras should have shown an intermediate spatio-temporal localization compared to the individual proteins, i.e. LATVav should have shown a pattern between the ones of LAT and Vav. Was this the case for all of the chimeras?

---

## [Author Response]

[…] The comments where new data appear to be necessary to address the concern are marked with asterisks (*).1) Regarding inconsistency in IL-2 protein and message:a) The data presented in Figure 1A and B are inconsistent. On the one hand, it is demonstrated that IL-2 amounts secreted to the supernatant of full stimulus T cells vs. full stimulus Itk KO at 10 µM is relatively unchanged, but strangely, on the other hand, the IL-2 mRNA is drastically lower in the Itk KO. No explanations are provided for this inconsistency between the protein and the mRNA levels. This issue should be addressed by the authors.* Could you provide a time course of IL-2 accumulation in comparison of IL-2 mRNA to provide a clear explanation and bolster the physiological significant of the subsequent investigation.

IL-2 mRNA and protein levels don’t necessarily align as IL-2 mRNA levels only measure IL-2 generation whereas the protein levels measure the difference between IL-2 generation and consumption. Concurrent diminished IL-2 generation and consumption can leave IL-2 protein levels largely unchanged while resulting in substantially reduced IL-2 mRNA levels.

We have provided a time course of IL-2 mRNA versus protein levels in Figure 1—figure supplement 1 showing that IL-2 generation is limited to the first few hours of the reactivation of primed T cells. Providing a molecular explanation for these dynamics we show that the transcription factor NFκB that is required for IL-2 gene transcription translocates to the nucleus with a comparable dynamics.

b) Figure 1C and D. The authors profile the signaling molecules at the different compartments of the IS at different settings, i.e., full stimulation in comparison with co-stimulation blockade (C) and full stimulation vs. full stimulus Itk deficiency (D). However, it is difficult to deduce clear-cut conclusions since the profile of signaling molecules in C is much wider than in D. The authors should explain why they neglect specific molecules in D that were analyzed in C. In addition, the authors should include the profile of signaling molecule localization for the combined treatment of co-stimulation blockade and Itk knockout, as these appear to create an additive effect. Similarly, for Figure 1—figure supplement 2B, Figure 1—figure supplement 3 and Figure 1—figure supplement 4, the graphs of full stimulation vs. co-stimulation blockade/Itk deficiency should compare similar signaling molecular kinetics. Also, Figure 1—figure supplement 3 shows the kinetics of Akt (co-stimulation blockade) that is missing in Figures 1—figure supplements 2 and 4 (Itk deficiency), and Figure 1—figure supplement 4 shows the kinetics of PKC that is missing in Figure 1—figure supplements 2 and 3.

The reviewers are correct in pointing out that we have not imaged all sensors equally across all experimental conditions. This has practical reasons. Our initial submission is based on the time-resolved image acquisition and analysis of 4781 cell couples in addition to previous data on more than a thousand of cell couples also included in Figure 1C/D. Extending the data to an equal coverage of sensors across all experimental conditions would have required more than a thousand additional cell couples. While we can efficiently handle large amounts of imaging data such extension still would require months of work. The imaging data as they are unambiguously support impaired formation of the cSMAC upon costimulation blockade and Itk deficiency. We agree with the reviewers that consistent sensor coverage across all experimental conditions is desirable and we are extending the Itk ko data set accordingly for future work. However, we believe that in the context of this manuscript the months of work required to harmonize the data sets doesn’t stand in a reasonable relation to the resulting gain in scientific insight.

In the supplementary figures we only show data that hasn’t been previously published with a reference to the relevant previous work in Figure 1—figure supplement 5. We will gladly add previously published work if requested.

c) In Figure 5, the authors should show IL-2 concentration in addition to IL-2 mRNA amounts due to the inconsistency between them in Figure 1. In addition, the authors show that LAT-V3 restored LAT centralization under co-stimulation inhibition/Itk deficiency, and that LAT-V3 partially restores Grb-2 and Lck centralization under Itk deficiency, yet it had no effect on IL-2 mRNA. This should be discussed by the authors, as it suggests that it is more than LAT centralization that affects IL-2 production.

We agree with the reviewers that it is interesting that the drastically enhanced central LAT-V3 localization with the accompanying moderate enhancement of Grb2 and Lck centrality in Itk-deficient T cells didn’t lead to restoration of IL-2 secretion. We had already discussed that our data strongly suggest that extent and dynamics of cSMAC formation, moderate and an emphasis on the first two minutes, respectively, are critical for its function in promotion IL-2 secretion with LAT-V3 getting both wrong. Very briefly, we suggest that excessive cSMAC formation as seen with LAT-V3 switches cSMAC function from a reaction crucible to sequestration of signaling intermediates. We now point out the delayed timing of the moderately enhanced Grb2 and Lck localization, have added data on enhanced Grb2 interface recruitment upon LAT-V3 expression in Itk-deficient T cells upon costimulation blockade and refer to an expanded Discussion section when showing the data.

As discussed in our response point 1a, mRNA generation is a more direct and more sensitive readout of T cell activation. Given the moderate changes in IL-2 mRNA levels shown, it seems highly unlikely that a determination of IL-2 protein levels by ELISA can contribute.

d) In Figure 5, the authors showed Grb-2, Lck, and VAV-1 centralization as a function of LAT-V3 presence upon stimulation, but omitted the LAT-VAV and LAT- PLCδPH constructs and how they affect signaling molecule centralization.* These should be included in the data as they also rescued LAT centralization. In addition, the authors should show how the different LAT constructs under the combined treatment of Itk and the co-stimulation blockade influence Grb-2, Lck, and VAV-1, as this may have an additive effect that can be noticeable.

We have exchanged a set of emails with Prof. Dustin regarding this concern.

Our question: We appreciate the importance of the question of how spatiotemporal signaling organization changes when we force central LAT localization. For that reason, we have included the Grb-2, Lck, Vav-1 localization data upon LAT-V3 expression. We will gladly extend these data to costimulation blockade in Itk-deficient T cells as suggested. However, recapitulating the entire LAT-V3 data with LAT-Vav and LAT- PLCδPH requires substantial effort well beyond two months. We first would have to clone the necessary six dual expression constructs and then image 24 experimental conditions (two times (LAT-Vav and LAT- PLCδPH) three (Grb-2, Vav-1, Lck) constructs times four stimuli) that with the somewhat lower efficiency of the dual expression constructs will require around 100 imaging runs. With work required to address other review concerns 4 imaging runs per week dedicated to this question seems reasonable. The LAT-V3 data already make the point that overall spatiotemporal changes upon forced LAT centralization are rather modest. While experiments obviously can yield intriguing surprises, it seems most likely that we will see similarly modest changes in spatiotemporal signaling organization upon expression of LAT-Vav and LAT-PLCδPH. I our opinion only investigating a larger number of sensors has a substantial chance of discovering meaningful differences in signaling organization upon expression of LAT-V3 versus LAT-Vav versus LAT-PLCδPH, something that is essentially a study in itself. Therefore, we are uncertain whether in an investigation of the localization of Grb-2, Lck, and Vav-1 upon expression of LAT-Vav and LAT-PLCδPH the effort required stands in a reasonable relation to the likely outcomes in the context of this manuscript. I would very much appreciate if you could suggest how you would like us to proceed.

The answer: “Thank you for calling our attention to this issue and please try address the reviewer concern in a manner that is feasible within the 2 month window. For example, an acceptable solution may be to include a fix and stain experiment to support the localisation of some of the endogenous players requested by the reviewer at key times predicted by your model. Please outline the difficulties in addressing the concern in your point-by-point response to the reviewer comments and this will allow us to take this into account when evaluating your revision.”

As requested by the reviewers, we have now imaged Grb2, Lck and Vav-1 in the presence of LAT V3 in Itk-deficient 5C.C7 T cells upon costimulation blockade. Interestingly, we found that expression of LAT V3 in Itk-deficient 5C.C7 T cells upon costimulation blockade triggered Grb2 interface accumulation to an extent not seen under any of the other experimental conditions. We suggest the following scenario to explain these findings. Grb2 doesn’t effectively compete for binding to the cSMAC under strong and even moderate T cell activation conditions. Likely, stronger binding signaling intermediates are activated to a sufficient extent under those conditions to outcompete Grb2. However, with the strong signaling attenuation provided by the combination of costimulation blockade and Itk-deficiency activation of more competitive signaling intermediates becomes sufficiently inefficient to allow Grb2 recruitment to the LAT V3-based signaling complexes.

We agree with the reviewers that an analysis of the spatiotemporal organization of T cell signaling upon expression of LAT-Vav and LAT-PLCδPH is of interest. We have, therefore, extensively thought about the feasibility of the staining experiments suggested in Prof. Dustin’s reply. Staining experiments actually require substantially larger numbers of T cells than our imaging experiments where 30,000 T cells are generally enough to complete an entire data set. For the experiments suggested here the T cells still need to be FACS sorted for expression of LAT-Vav or LAT-PLCδPH and T cell numbers thus can become limiting. Moreover, we know from the careful analysis of T cell morphology in electron microscopy experiments (Roybal et al., 2015) that even when using centrifugation to force cell coupling, fixed T cell/APC samples invariably contain a range of time points relative to tight cell couple formation that can only be approximated. Given the importance of cSMAC dynamics established in our manuscript imprecise determination of time of cell coupling is highly problematic. Looking at the 12 conditions imaged with LAT-V3, there is no obvious condition that when determined in isolation should provide substantial insight into how LAT-Vav or LAT-PLCδPH shape the spatiotemporal organization of T cell signaling. This leaves us with having to use staining experiments with the constraints just discussed across a range of T cell activation conditions, something likely to take months of work. We thus believe that any meaningful insight into the spatiotemporal organization of T cell signaling upon expression of LAT-Vav and LAT-PLCδPH remains well beyond the (already extended) time scale of this revision. We don’t believe that lack of such data weakens the manuscript: As overexpression of the more dominantly localized LAT-V3 had largely minor impacts on the localization of Grb2, Lck and Vav-1, is seems unlikely that the less dominantly localized LAT-Vav and LAT-PLCδPH will yield changes that are sufficientlydramatic to be readily detectable.

Despite these concerns we have executed a first staining experiment as suggested. We have sorted T cells transduced to express LAT-Vav-GFP, activated them with APC and peptide for 1min under full stimulus and costimulation-blocked conditions, fixed and stained the cell couples with an antibody against active Src kinase family members as shown in Author response image 1.

**Author response image 1. respfig1:** Representative anti-phospho Src staining experiment. Given are 3 5C.C7/LAT-Vav-GFP T cell CH27 B cell APC couples after one minute of interaction under full stimulus conditions as fixed and stained with anti-phospho Src 419-Alexa 568 and imaged by confocal microscopy. Limiting LAT-Vav-GFP fluorescence and lack of central phospho Src kinase family staining are evident.

Limiting the biological information that could be gained from this experiment, we, first, lost part of the GFP-fluorescence during cell couple fixation. A reliable determination of the formation of a LAT-Vav-based cSMAC became questionable. Second, as Src family kinases are expressed in both the T cell and the APC, unambiguous assignment of interface fluorescence to either cell type was not feasible. This limitation will similarly extend to a determination of interface accumulation of Vav-1 or Grb2. Even assuming dominant accumulation of active Src kinase family members in the T cell, substantial central Src kinase family activation was not found. With all technical caveats to be considered, these data seem consistent with the limited central accumulation of Lck upon expression of LAT-V3 (Figure 6). We believe that these experiments illustrate our willingness to pursue reviewers’ suggestions and support our notion that fixed cell staining experiments are unlikely to generate insight into the organization of T cell signaling upon cSMAC reconstitution any more efficiently than the live cell imaging experiments do.

2) Missing information: a) * In the WBs of LAT phosphorylation (Figure 2—figure supplement 2 and Figure 4—figure supplement 1F) the unstimulated time point (often called 0 min stimulation) is missing. Thus, the fold increase upon stimulation remains unknown. Similarly, the total amount of LAT needs to be shown; e.g. in S5D GFP-LATV3 is less phosphorylated than wt GFP-LAT. Was this due to lower expression levels of LATV3 and per molecule there was actually more LATV3 phosphorylation than wt LAT phosphorylation?

We have included a no peptide control in the first three of the seven experimental replicates of the determination of LAT phosphorylation as a function of time. Having established minimal LAT phosphorylation as already published twice before using the same experimental system (Purtic et al., 2005, Supplementary Figure 1 and Singleton et al., 2006,, Figure 8B), we then dropped this control for the remainder of the experiments. We apologize that the level of LAT phosphorylation upon no peptide stimulation was shown in a rather misleading fashion in our original submission. We have now revised Figure 2D to more clearly identify this important control. The representative Western blots shown in the supplement came from repeats without the no peptide control. Author response image 2 shows one of the earlier blots (lacking the 10 min time point, without Itk-deficient T cells, and with an additional experimental condition not further pursued) demonstrating the lack of LAT phosphorylation in the absence of agonist peptide (lane labelled ‘-‘). We will gladly add the blot to the supplement if requested.

**Author response image 2. respfig2:** anti-phospho LAT Western blot with no peptide control. 5C.C7 T cells were activated with CH27 B cell APCs for the indicated time in minutes under full stimulus (‘wt’) and costimulation blocked (‘αB7’) conditions, lysed and blotted with an anti-LAT pY191 antibody. T cells were treated with pervanadate as a positive control (‘+’) and activated with CH27 B cell APCs in the absence of MCC agonist peptide as a negative control (‘–‘). The ‘WT+P’ conditions include treatment of T cells with a protein transduction reagent and have not been further pursued as part of this manuscript.

The expression level of all wild type and mutant molecules was fixed by FACS sorting on the GFP tag and is, therefore, the same across the entire manuscript with a concentration of about 2µM (Sci. Signal. 2009, 2, ra15). This is now stated consistently. For further quantitative insight, we have now twice measured the amount of GFP-tagged LAT relative to the endogenous protein and report this quantification in the Results section and Figure 4—figure supplement 1B, C. Briefly, GFP fusion proteins were expressed at 2.1 ± 0.7-fold the endogenous level of LAT in non-transduced T cells (Figure 4—figure supplement 1B, C) with little change to endogenous LAT levels in the transduced T cells. We will gladly add a third replicate if requested. This quantification extends equally to all LAT fusion proteins because of the use of GFP-based FACS sorting.

b) * Relative expression levels of wt to mutant molecules need to be shown for the most important experiments.

The expression of all wild type and mutant molecules was fixed by FACS sorting on the GFP tag. This is now stated consistently.

c) It’s not exactly clear how the authors go from the STED results to the volumes. In order to be able to do this the authors would need to report the resolution of the STED system. Is resolution with STED will have a tradeoff between lateral and axial resolution the best isotropic resolution for 3D step might be on the order of 80 nm xy and 100 nm z. Higher xy resolution might be achieved, but only at cost of degraded z resolution.

We have measured the closest detectable distance between adjacent puncta as the experimentally achieved resolution as 100nm in all dimensions. This results in a smallest theoretically detectable cluster size of 0.001µm^3^, well below the experimental cut-off of 0.04/0.06µm^3^ that distinguishes non-specific from LAT/phospho-LAT-specific signals, as detailed in the Materials and methods section. We now report the experimentally determined STED resolution in the Materials and methods.

d) The correlative fluorescence-EM seems to be focused on temporal correlation but it’s not clear if they are look at the same call imaged by fluorescence and if they can register the fluorescence and EM data? Can they really say what structure is the fluorescence reported cSMAC in a given EM view. * A supplementary figure could be constructed to better illustrate how and to what level they are look at correlations.We have now generated a figure to illustrate the correlative light electron microscopy work flow, Figure 3—figure supplement 1. Because of the use of finder grids we can readily match cells in the bright field and EM images. Because of the resolution limits of spinning disk confocal microscopy, a sub µm-scale registration of the fluorescence and electron microscopy data is difficult. To account for this difficulty, we have chosen a conservative image analysis approach that biases the outcome against the hypothesis to be tested, i.e. that enhanced membrane undulations are associated with the cSMAC: To determine membrane undulations in the cSMAC region, the interface diameter in the electron micrograph was divided into four equal sections with the central two sections defined as the cSMAC. This is a conservative assumption as cSMACs were by definition contained within this region, yet commonly smaller than the entire half of the interface diameter. This is now detailed in the Materials and methods section.3) Chimeric receptors:a) Figure 3. The authors should introduce the fused V3 domain to LAT when it is first used, and present the goal of this experiment. I assume that the authors used LAT-V3 under co-stimulation blockade to demonstrate that it rescue membrane undulation; however, this is not mentioned. Did co-stimulation blockade abrogate membrane undulation in Figure 3? Figure 3D does not show these data. Although LAT-V3 is mentioned in Figure 3, it is explained only in Figure 4.

We now briefly introduce the LAT-V3 construct in the description of the CLEM experiments and refer to the LAT-V3 schematic in the legend of Figure 3C. However, to avoid duplication, we leave a full justification of the use LAT-V3 to the detailed Introduction in the subsequent paragraph as referred to now.

We have earlier published reduced membrane undulations upon costimulation blockade and now refer to these data. That said, the aim of the CLEM experiments was to determine whether cSMAC formation is associated with enhanced membrane undulations. Costimulation blockade substantially impairs cSMAC formation, making the determination of an association of cSMAC formation with membrane undulation rather difficult. Yet published data are now referred to. Activation of 5C.C7 T cells expressing LAT-V3 under full stimulus conditions as another possible comparator leads to enhancement of cSMAC formation very similar to the LAT-V3/costimulation blocked condition. We are not sure what additional insight would be gained including both conditions. Having to decide between investigating LAT-V3 under full stimulus or costimulation-blocked conditions we felt that a restorative condition, i.e. a condition with cSMAC formation forced when normally absent, would be more informative.

b) * Equipping LAT (but also SLP76 and Grb2) with new interaction domains is supposed to generate chimeric molecules that bind to new binding partners. For example, LATVav should also bind to proteins that normally would bind to Vav. This should be tested by classical immunoprecipitation experiments (and if this is difficult due to low cell numbers in the primary cells, one could do it with stable cell lines expressing the chimeric molecules). Without these experiments the new interactions remain hypothetical.

We agree with the reviewers that a determination of the changes in the LAT interactome upon addition of new protein interaction domains is of interest. We have tried to address this question using pulldowns with anti-GFP beads. These experiments proved complex in having to consider number of T cells required, T cell activation conditions (resting, pervandate, APC plus peptide) and means of analysis of LAT binders (hypothesis-driven or unbiased, background of unspecific bead-binding proteins). We therefore exchanged emails with the editor regarding these challenges and received the following answer:

“If the authors have tried the biochemistry and failed on this, the best option may just be to acknowledge this in the paper and that the interactions are predicted, but are difficult to verify. We know that all of the individual SH2 and SH3 interactions are very weak and could not lead to immunoprecipitation on their own. Extracting complexes that are stable enough for indirect immunoprecipitation depends upon multi protein interactions that may be hard to predict..… So if the manuscript is otherwise ready I think they should acknowledge that these interactions are predicted, but were not sufficiently stable to be captured by immunoprecipitation and resubmit.”

We have now acknowledged in the Results section that ‘while the addition of protein interaction domains to LAT is predicted to alter their interactome, initial experiments to support this notion remained inconclusive’.

c) The chimeras should have shown an intermediate spatio-temporal localization compared to the individual proteins, i.e. LATVav should have shown a pattern between the ones of LAT and Vav. Was this the case for all of the chimeras?

In the context of supramolecular complex formation the localization of a fusion protein is not the ‘average’ of the localization of its components. For example, the PKCθ V3 domain in isolation does not localize to the interface center even under full stimulus conditions (Figure 5—figure supplement 1A) nor does LAT upon costimulation blockade (Figure 2B). However, the LAT-V3 fusion protein shows dominant central localization upon costimulation blockade (Figure 4B). How is this possible? By generating a LAT-V3 fusion proteins one doesn’t only add the localization preferences of LAT and the PKC V3 domain, one also increases the valence of the fusion protein over its components. As increased valence is a key facilitator of recruitment to supramolecular complexes localization properties of fusion proteins can become qualitatively different from those of their components. This exciting argument has now been added to the Discussion section.